# Sparse Spiking Neural Network: Exploiting Heterogeneity in Timescales for Pruning Recurrent SNN

**Biswadeep Chakraborty, Beomseok Kang, Harshit Kumar & Saibal Mukhopadhyay**
Department of Electrical and Computer Engineering
Georgia Institute of Technology
Atlanta, GA 30332, USA
`{biswadeep, smukhopadhyay6}@gatech.edu`

## Abstract

Recurrent Spiking Neural Networks (RSNNs) have emerged as a computationally efficient and brain-inspired learning model. The design of sparse RSNNs with fewer neurons and synapses helps reduce the computational complexity of RSNNs. Traditionally, sparse SNNs are obtained by first training a dense and complex SNN for a target task, and, then, pruning neurons with low activity (activity-based pruning) while maintaining task performance. In contrast, this paper presents a task-agnostic methodology for designing sparse RSNNs by pruning a large randomly initialized model. We introduce a novel Lyapunov Noise Pruning (LNP) algorithm that uses graph sparsification methods and utilizes Lyapunov exponents to design a stable sparse RSNN from a randomly initialized RSNN. We show that the LNP can leverage diversity in neuronal timescales to design a sparse Heterogeneous RSNN (HRSNN). Further, we show that the same sparse HRSNN model can be trained for different tasks, such as image classification and temporal prediction. We experimentally show that, in spite of being task-agnostic, LNP increases computational efficiency (fewer neurons and synapses) and prediction performance of RSNNs compared to traditional activity-based pruning of trained dense models.

## 1 Introduction

Recurrent Spiking Neural Networks (RSNNs), inspired by the human brain's information processing mechanism, utilize spikes for efficient learning and processing of spatio-temporal data (Maass, 1997). Recent advancements in SNN research have underscored the importance of leveraging heterogeneity in neuronal parameters to optimize network performance (Chakraborty & Mukhopadhyay, 2023b; Perez-Nieves et al., 2021; She et al., 2021). These studies have demonstrated that RSNNs with diversity in neurons' integration/relaxation dynamics, referred to as the heterogeneous RSNN (HRSNN), enhance the learning ability of RSNNs and show improved performance over homogeneous spiking neural networks in tasks such as spatio-temporal classification of video activity recognition (Chakraborty & Mukhopadhyay, 2022; 2023a; Chakraborty et al., 2023; Padmanabhan & Urban, 2010).

Though the introduction of such heterogeneity in the neuronal parameters helps in improving the performance of the model, it also increases the complexity of the model exponentially, especially as the number of neurons increases. This makes optimizing the model hyperparameters extremely hard. Moreover, standard sparse random initializations of the network make it very unstable, as observed from their Lyapunov spectra [See Suppl. Sec. A.7] . The design of sparse HRSNN models with fewer neurons and synapses helps balance computational demand and performance. Thus, getting a sparse HRSNN model without sacrificing on the performance is of utmost importance. Traditionally, sparse neural networks are designed by first training a dense (complex) network for a target task, followed by pruning neurons/synapses to reduce computation while minimizing performance drop for that task (Blalock et al., 2020; Chowdhury et al., 2021b; Chen et al., 2018). Various task-dependent pruning methods have been explored for feed-forward SNNs. For example, STDP-based pruning of connections and weight quantization of SNNs have been studied for energy-efficient recognition

(Rathi et al., 2018). Likewise, the lottery ticket hypothesis has been studied for complex (large) SNNs (Kim et al., 2022b). Gradient rewiring has also been explored for pruning deep SNNs (Chen et al., 2021). However, the pruned models derived from such task-driven pruning algorithms are extremely overfitted to the task it is trained on and demonstrate poor generalization performance.[See Suppl. Sect. B.3]]. Also, since most of these methods are mostly adapted from pruning techniques of deep neural networks (DNNs), they do not leverage the unique temporal dynamics and neuronal timescales inherent to heterogeneous spiking networks. Further, as these methods consider performance on a target task during pruning, the complexity of the final pruned models varies from task to task and even across datasets for a given task.

In this paper, we present a novel **task-agnostic method**, referred to as Lyapunov Noise Pruning (LNP), for designing sparse HRSNN. In contrast to the conventional approach of designing sparse networks by task-dependent pruning of trained dense networks, our approach starts with a randomly initialized and arbitrarily initialized dense (complex) HRSNN model. We leverage the Lyapunov spectrum of an HRSNN model and techniques from spectral graph sparsification algorithms (Spielman & Srivastava, 2011; Moore & Chaudhuri, 2020) to prune synapses and neurons while keeping the network stable (Spielman & Srivastava, 2011; Moore & Chaudhuri, 2020; Vogt et al., 2020). The resulting random sparse HRSNN can next be trained for different target tasks using supervised (backpropagation) or unsupervised (Spike-Time-Dependent-Plasticity, STDP) methods.

Our **task-agnostic sparse model design** helps develop universally robust and adaptable models and eliminates the need for extensive task-specific adjustments (You et al., 2022; Liu et al., 2022b). Instead of minimizing (or constraining) performance loss for a given task, LNP optimizes the model structure and parameters while pruning to preserve the stability of the sparse HRSNN. This results in sparse models that are stable and flexible, and maintain performance across multiple tasks. We further show that the same sparse HRSNN obtained by LNP can be trained for various tasks, namely, image classification and time-series prediction. For image classification on the CIFAR10, CIFAR100 datasets, the sparse HRSNN from LNP shows similar performance but at a much lower (as both neurons and synapses are pruned) computation cost. Likewise, we assess the efficacy of the proposed task-agnostic pruning approach by training the sparse HRSNN models for prediction tasks. We consider (1) synthetic datasets of chaotic systems, such as Lorenz and Rossler, and (2) real-world datasets, including Google Stock Price Prediction and Wind Speed Prediction. The experimental results show that the proposed LNP enhances the pruning efficiency (defined as the ratio of performance to synaptic operations) over conventional task-dependent activity-based pruning methods. The key contributions of this paper are as follows:

- **Sparse Recurrent Spiking Network Design Methods.** We present a methodology for designing sparse recurrent spiking networks by task-agnostic pruning of a randomly initialized dense model. This is in contrast to prior SNN pruning approaches that are task-dependent, designed for feed-forward SNNs, and adapted from DNN pruning methods thereby ignoring unique temporal dynamics of recurrent spiking networks.

- **Lyapunov Noise Pruning Algorithm.** We present a novel Lyapunov-based noise pruning algorithm, using spectral graph sparsification methods and the Lyapunov spectrum of a HRSNN network. The proposed algorithm eliminates neurons and synapses from a randomly initialized dense HRSNN model by preserving the delocalized eigenvectors for improved stability and performance of the resulting sparse HRSNN, without training it on any dataset.

- **Effective Utilization of Neuronal Timescale Heterogeneity.** The proposed approach leverages the diversity of neuronal timescales in an HRSNN to assist in pruning and enhance the performance of the sparse HRSNN.

- **Task-agnostic Sparse Model Design.** The pruning algorithms developed in this paper is task-agnostic (unsupervised). The proposed approach do not optimize for performance on a given dataset while pruning; rather, only optimizes the topology of the graph and the neuronal time constants of the HRSNN network while ensuring stability of the network.

## 2    PRELIMINARIES AND DEFINITIONS

**Models:** For this paper, we use a Leaky Integrate and Fire (LIF) neuron model. The heterogeneity is introduced by assigning distinct membrane time constants, $\tau_{m,i}$, to each LIF neuron, creating a varied

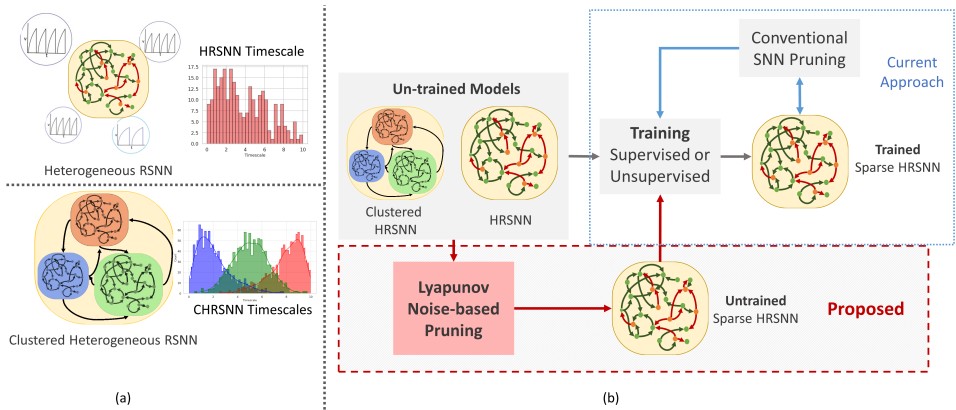

Figure 1: (a) Concept of HRSNN with variable Neuronal and Synaptic Dynamics (b) Figure showing the task-agnostic pruning and training of the CHRSNN/HRSNN networks using LNP in comparison to the current approach

distribution of these parameters. Using the above notion of heterogeneous LIF neurons, we define two distinct HRSNN models, presented in Fig 1(a): First, the **HRSNN** model, predicated on the framework introduced in recent research (Chakraborty & Mukhopadhyay, 2023b; 2022), where the time constants of each neuron are derived by sampling from a gamma distribution, engendering heterogeneity in neuronal dynamics. Again, recent advancements in neuroscience have also elucidated that the human brain comprises multiple regions, each exhibiting distinct temporal properties, conceptualized as the multi-region network model (Perich et al., 2020; de Oliveira Junior et al., 2022). Inspired by these, we propose the **Clustered Heterogeneous RSNN (CHRSNN)** model, wherein the recurrent layer comprises multiple clusters of HRSNN models, each characterized by a unique distribution of time constants. Each of these neurons is then connected randomly to form a small-world network architecture of the overall CHRSNN.

---

**Algorithm 1** Lyapunov Noise Pruning (LNP) Method

---

1: **Step 1: Synapse Pruning using Spectral Graph Pruning**
2: **for** each $e_{ij}$ in $A$ connecting $n_i, n_j$ **do**
3:     Find $\mathcal{N}(e_{ij})$
4:     Compute $\lambda_k$ for $k$ in $\mathcal{N}(e_{ij})$ (Algorithm 2)
5:     Define $W$ using harmonic mean$(\lambda_k)$ for $k$ in $\mathcal{N}(e_{ij})$ (Eq 1)
6:     Use $\boldsymbol{b}(t)$ at each node
7:     Compute $C$ of firing rates
8:     Preserve each $e_{ij}$ with $p_{ij}$ yielding $A^{\text{sparse}}$ for $i \neq j$
9: **end for**
10: **Step 2: Node Pruning using Betweenness Centrality**
11: **for** each $n_i$ in Network **do**
12:     Compute $B(n_i)$ (Algorithm 3)
13:     **if** $B(n_i) <$ threshold **then**
14:         Remove $n_i$
15:     **end if**
16: **end for**
17: **Step 3: Delocalization of the Eigenvectors**
18: **for** each $A^{\text{pruned}}$ **do**
19:     Add edges to preserve eigenvectors and maintain stability
20: **end for**
21: **Step 4: Neuronal Timescale Optimization**
22: **for** each pruned model $m$ **do**
23:     Use Lyapunov spectrum $L$ to optimize neuronal timescales $\tau_i \forall i \in \mathcal{R}$ using BO.
24: **end for**
25: **return** $A^{\text{pruned}} = 0$

---

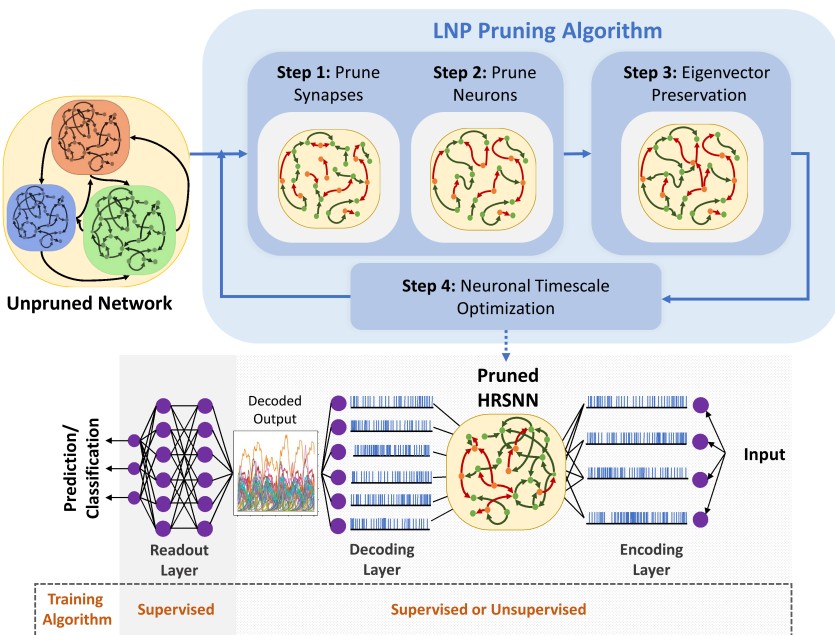

Figure 2: Complete flowchart showing the steps for the LNP pruning algorithm and the training methodology to use the pruned HRSNN network

**Methods: 1.** *Lyapunov Spectrum of RNN* using the algorithm from (Vogt et al., 2020) by observing the network's contraction/expansion over time sequences. The details are discussed in Suppl. Sec. A and Algorithm2.

**2.** *Spectral Graph Sparsification Methods* (Feng, 2019; Liu & Yu, 2023) are crucial for simplifying graphs while maintaining their fundamental structural and spectral attributes. They seek ultra-sparse subgraphs that effectively approximate the original Laplacian eigenvalues and eigenvectors.

**3.** *Betweenness Centrality:* $(C_b)$ In a network with recurrent connections, $C_b$ gauges a node's importance in facilitating communication between different network sections. Nodes with high $C_b$ form critical information channels connecting various parts of the network.

## 3 METHODS

### 3.1 LYAPUNOV NOISE PRUNING (LNP) METHOD

The proposed research presents a pruning algorithm employing spectral graph pruning and Lyapunov exponents in an unsupervised model. We calculate the Lyapunov matrix, optimizing for ratios and rates to handle extreme data values and incorporate all observations. After pruning, nodes with the lowest betweenness centrality are removed to improve network efficiency, and select new edges are added during the delocalization phase to maintain stability and integrity. This method balances structural integrity with computational efficacy, contributing to advancements in network optimization. Algorithm 1 shows a high-level algorithm for the entire process.

**Step I: Noise-Pruning of Synapses:** First, we define the Lyapunov matrix of the network. To formalize the concept of the Lyapunov Matrix, let us consider a network represented by a graph $G(V, E)$ where $V$ is the set of nodes and $E$ is the set of edges. For each edge $e_{ij}$ connecting nodes $z$, let $N(z)$ be the set of neighbors of nodes $z, z = \{i, j\}$. Thus, the Lyapunov exponents corresponding to these neighbors are represented as $\Lambda(N(z))$. The element $L_{ij}$ of the Lyapunov Matrix $(L)$ is then calculated using the harmonic mean of the Lyapunov exponents of the neighbors of nodes $i, j$ as:

$$L_{ij} = \frac{n \cdot |\Lambda(N(i)) \cup \Lambda(N(j))|}{\sum_{\lambda \epsilon \Lambda(N(i)) \cup \Lambda(N(j))} \frac{1}{\lambda}} \quad (1)$$

where $\lambda$ denotes individual Lyapunov exponents from the set of all such exponents of nodes $i$ and $j$'s neighbors. $L$, encapsulates the impact of neighboring nodes on each edge regarding their Lyapunov exponents, which helps us evaluate the network's stability and dynamical behavior. Through the harmonic mean, the matrix accommodates the influence of all neighbors, including those with extreme Lyapunov exponents, for a balanced depiction of local dynamics around each edge. Thus, the linearized network around criticality is represented as:

$$\dot{\boldsymbol{x}} = -D\boldsymbol{x} + L\boldsymbol{x} + \boldsymbol{b}(t) = A\boldsymbol{x} + \boldsymbol{b}(t) \tag{2}$$

Here $\boldsymbol{x}$ represents the firing rate of $N$ neurons, with $x_i$ specifying the firing rate of neuron $i$. $\boldsymbol{b}(t)$ denotes external input, including biases, and $L$ is the previously defined Lyapunov matrix between neurons. $D$ is a diagonal matrix indicating neurons' intrinsic leak or excitability, and $A$ is defined as $A = -D + L$. The intrinsic leak/excitability, $D_{ii}$, quantifies how the firing rate, $x_i$, of neuron $i$ alters without external input or interaction, impacting the neural network's overall dynamics along with $\boldsymbol{x}$, $L$, and external inputs $\boldsymbol{b}(t)$. Positive $D_{ii}$ suggests increased excitability and firing rate, while negative values indicate reduced neuron activity over time. We aim to create a sparse network ($A^{\text{sparse}}$) with fewer edges while maintaining dynamics similar to the original network. The sparse network is thus represented as:

$$\dot{\boldsymbol{x}} = A^{\text{sparse}}\boldsymbol{x} + \boldsymbol{b}(t) \quad \text{such that} \quad \left| \boldsymbol{x}^T \left( A^{\text{sparse}} - A \right) \boldsymbol{x} \right| \le \epsilon \left| \boldsymbol{x}^T A \boldsymbol{x} \right| \quad \forall \boldsymbol{x} \in \mathbb{R}^N \tag{3}$$

for some small $\epsilon > 0$. When the network in Eq. 6 is driven by independent noise at each node, we define $\boldsymbol{b}(t) = \boldsymbol{b} + \sigma\boldsymbol{\xi}(t)$, where $\boldsymbol{b}$ is a constant input vector, $\boldsymbol{\xi}$ is a vector of IID Gaussian white noise, and $\sigma$ is the noise standard deviation. Let $\Sigma$ be the covariance matrix of the firing rates in response to this input. The probability $p_{ij}$ for the synapse from neuron $j$ to neuron $i$ with the Lyapunov exponent $l_{ij}$ is defined as:

$$p_{ij} = \begin{cases} \rho l_{ij}(\Sigma ii + \Sigma jj - 2\Sigma ij) & \text{for } w_{ij} > 0 \text{ (excitatory)} \\ \rho |l_{ij}|(\Sigma ii + \Sigma jj + 2\Sigma ij) & \text{for } w_{ij} < 0 \text{ (inhibitory)} \end{cases} \tag{4}$$

Here, $\rho$ determines the density of the pruned network. The pruning process independently preserves each edge with probability $p_{ij}$, yielding $A^{\text{sparse}}$, where $A_{ij}^{\text{sparse}} = A_{ij}/p_{ij}$, with probability $p_{ij}$ and $0$ otherwise. For the diagonal elements, denoted as $A_{ii}^{\text{sparse}}$, representing leak/excitability, we either retain the original diagonal, setting $A_{ii}^{\text{sparse}} = A_{ii}$, or we introduce a perturbation, $\Delta_i$, defined as the difference in total input to neuron $i$, and adjust the diagonal as $A_{ii}^{\text{sparse}} = A_{ii} - \Delta_i$. Specifically, $\Delta_i = \sum_{j \ne i} |A_{ij}^{\text{sparse}}| - \sum_{j \ne i} |A_{ij}|$. This perturbation, $\Delta_i$, is typically minimal with a zero mean and is interpreted biologically as a modification in the excitability of neuron $i$ due to alterations in total input, aligning with the known homeostatic regulation of excitability.

**Step II: Node Pruning based on Betweenness Centrality:** In addressing network optimization, we propose an algorithm specifically designed to prune nodes with the lowest betweenness centrality ($C_b$) in a given graph, thereby refining the graph to its most influential components. $C_b$ quantifies the influence a node has on information flow within the network. Thus, we calculate $C_b$ for each node and prune the nodes with the least values below a given threshold, ensuring the retention of nodes integral to the network's structural and functional integrity. The complete algorithm for the node pruning is given in Algorithm 3 in Suppl. Sect. E.

**Step III: Delocalizing Eigenvectors:** To preserve eigenvector delocalization and enhance long-term prediction performance post-pruning, we introduce a predetermined number of additional edges to counteract eigenvalue localization due to network disconnection. Let $G = (V, E)$ and $G' = (V, E')$ represent the original and pruned graphs, respectively, where $E' \subset E$. We introduce additional edges, $E''$, to maximize degree heterogeneity, $H$, defined as the variance of the degree distribution $H = \frac{1}{|V|} \sum_{v \in V} (d(v) - \bar{d})^2$, subject to $|E' \cup E''| \le L$, where $L$ is a predetermined limit. This is formalized as an optimization problem to find the optimal set of additional edges, $E''$, enhancing eigenvector delocalization and improving the pruned network's predictive performance. Given graph $G = (V, E)$, with adjacency matrix $A$ and Laplacian matrix $L$, we analyze the eigenvalues and eigenvectors of $L$ to study eigenvector localization. To counteract localization due to pruning, we introduce a fixed number, $m$, of additional edges to maximize the variance of the degree distribution, $\text{Var}(D)$, within local neighborhoods, formalized as:

$$\max_{E''} \quad \text{Var}(D) \quad \text{subject to} \quad |E''| = m, \quad E' \cap E'' = \varnothing, \quad E'' \subseteq \bigcup_{v_i \in V} N(v_i) \times \{v_i\} \tag{5}$$

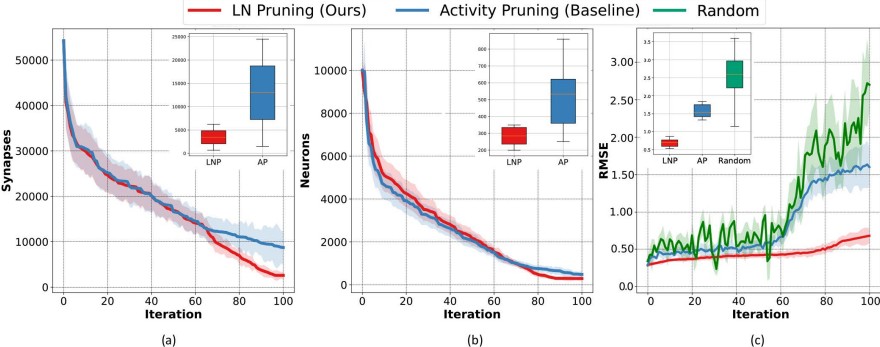

Figure 3: Comparative Evaluation of Pruning Methods Across Iterations. Figs. (a) and (b) show the evolution of the number of synapses and neurons with the iterations of the LNP and AP algorithms. Fig (c) represents how the RMSE loss changes when the pruned model after each iteration is trained and tested on the Lorenz63 dataset

This approach ensures eigenvector delocalization while preserving structural integrity and specified sparsity, optimizing the model's long-term predictive performance.

The goal is to maximize the variance of the degree distribution, $\text{Var}(D)$, by selecting the best set of additional edges $E''$ such that: (i) The number of additional edges is $m$. (ii) The additional edges are not part of the original edge set $E'$. (iii) The additional edges are selected from the neighborhoods of the vertices.

**Step IV: Neuronal Timescale Optimization** In optimizing RSNN, known for their complex dynamical nature, we employ the Lyapunov spectrum to refine neuronal timescales using Bayesian optimization. During optimization, RSNNs are inherently unstable due to their variable parameters and pruning processes, affecting training dynamics' stability and model learnability. Lyapunov exponents, as outlined in (Vogt et al., 2020), are crucial for understanding system stability and are linked to the operational efficiency of RNNs. This paper utilizes the Lyapunov spectrum as a criterion for optimizing neuronal timescales in pruned networks, aiming to maintain stability and functionality while minimizing instability risks inherent in pruning. Further details on Bayesian optimization and finalized timescales are in Suppl. Secs. D, B respectively.

# 4 EXPERIMENTS AND RESULTS

## 4.1 EXPERIMENTAL SETUP

The experimental process, depicted in Fig. 2, begins with a randomly initialized HRSNN and CHRSNN. Pruning algorithms are used to create a sparse network. Each iteration of pruning results in a sparse network; we experiment with 100 iterations of pruning. We characterize the neuron and synaptic distributions of the "Sparse HRSNN" obtained after each pruning iteration to track the reduction of the complexity of the models with pruning.

We train the sparse HRSNN model obtained after each pruning iteration to estimate performance on various tasks. Note, **the pruning process does not consider the trained model or its performance during iterations**. As outlined in Fig. 2, the sparse HRSNNs are trained for time-series prediction and image classification tasks. For the prediction task, the network is trained using 500 timesteps of the datasets and is subsequently used to predict the following 100 timesteps. For the classification task, each input image was fed to the input of the network for Tinput = 100 ms of simulation time in the form of Poisson-distributed spike trains with firing rates proportional to the intensity of the pixels of the input images, followed by 100 ms of the empty signal to allow the current and activity of neurons to decrease. For both tasks, the input data is converted into spike trains via rate-encoding, forming the high-dimensional input to the XRSNN. The output spike trains are then processed through a decoder and readout layer for final predictions or classification results.

**Evolution of Complexity of Sparse Models during Pruning:** We plot the change in the number of neurons and synapses for the 100 iteration steps for AP and LNP Pruning algorithms. The results for the variation of the synapses and neurons with the iterations of the pruning algorithm are plotted in Figs. 3(a) and (b), respectively. The LNP methods perform better than the activity-based pruning method and converge to a model with fewer neurons and synapses. Also, the inset diagram shows the variance of the distribution as we repeat the experiment 10 times for each algorithm. We added the results for random initialization for each step of the iteration of the LNP - initialized 10 different networks, trained them, and showed their performance in 3(c). After each iteration, we randomly created a network with an equal number of synapses and edges as found by the LNP method and trained and tested it on the Lorenz63 dataset to get the RMSE loss. We see that the Random initialized network shows higher variance and shows more jumps signifying the randomly initialized model is unstable without proper finetuning. In addition, we see the performance consistently getting

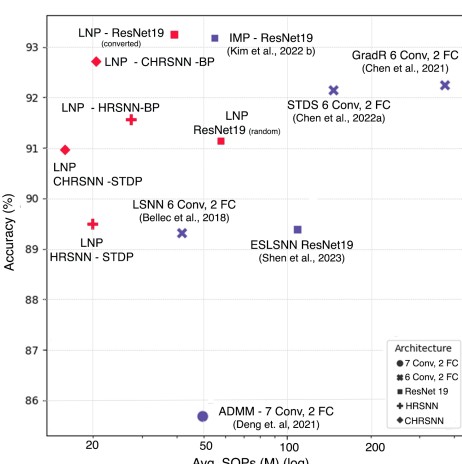

Figure 4: Scatter Plot showing Accuracy vs. Avg. SOPs for different pruning methods on CIFAR10. Results for CIFAR100 & Lorenz63 are given in Suppl. Sec. B.4

worse as the model size keeps decreasing, signifying an optimal network architecture is more crucial for smaller networks than for larger networks. We see that the final distribution for both synapses and neurons of the AP-based models has a higher variance than the LNP algorithm. This also highlights the stability of the proposed LNP algorithm. In addition to this, we also plot the final distributions of the timescales observed from the two methods. The complete results are shown in Suppl. Sec. B.

**Datasets:** We evaluate the performance of the LNP pruning methods for (1) time-series prediction on chaotic systems (i.e., Lorenz Xu et al. (2018a) and Rossler systems Xu et al. (2018b)) and two real-world datasets - Google Stock Price Prediction and wind speed prediction datasets Samanta et al. (2020) and (2) image classification on CIFAR10 and CIFAR100 datasets Krizhevsky et al. (2009). Further details are given in Suppl. Sec. A.

**Baselines:** We use the activity-based pruning (AP pruning) method Rust[1] et al. (1997) as the baseline, where we prune the neurons with the lowest activations in each iteration. The details of the AP pruning are given in Algorithm 4 and Suppl. Sec. A. This algorithm operates iteratively, pruning the least active neurons and retraining the model in each iteration. We also compare the LNP algorithm with current task-dependent state-of-the-art pruning algorithms from prior works, and the results are shown in Table 1. In addition, we also introduce the Random initialization method, where after each iteration of the LNP algorithm, we observe the number of neurons and synapses of the LNP pruned network and then generate a random Erdos-Renyi graph with the same number of neurons and synapses. We repeat this process for each step of the iteration. The results of the Random initialization method are shown in Fig. 3(c).

**Evaluation Metric** First, we use the standard RMSE loss which is given as $\text{RMSE}(t) = \sqrt{\frac{1}{D}\sum_{i=1}^{D}\left[\frac{u_i^f(t) - u_i(t)}{\sigma_i}\right]^2}$ where $D$ is the system dimension, $\sigma$ is the long term standard deviation of the time series, $\epsilon$ is an arbitrary threshold, and $u^f$ is the forecast. We also use another measure to measure the performance of prediction called valid prediction time (VPT) (Vlachas et al., 2020). The VPT is the time $t$ when the accuracy of the forecast exceeds a given threshold. Thus $VPT(t) = \sum \mathbb{I}(RMSE(t) < \epsilon)$ For these experiments, we set $\epsilon$ arbitrarily to 0.1. Thus, a higher VPT indicates a better prediction model.

**Energy Efficiency:** In assessing the energy consumption of neuromorphic chips, the central measure is the energy required for a single spike to pass through a synapse, a notably energy-intensive process Furber (2016). The overall energy consumption of a Spiking Neural Network (SNN) can be estimated by counting the synaptic operations (SOPs), analogous to floating-point operations (FLOPs) in

Table 1: Comparison of Pruning Methods(*=CIFAR10 pruned model trained & tested on CIFAR100)

| Method | Spiking Model | CIFAR10 | | | | | CIFAR100 | | | | |
|---|---|---|---|---|---|---|---|---|---|---|---|
| | | Baseline Accuracy | Accuracy Loss | Neuron Sparsity | Synapse Sparsity | SOP Ratio | Baseline Accuracy | Accuracy Loss | Neuron Sparsity | Synapse Sparsity | SOP Ratio |
| ADMM *Deng et al. (2021)* | 7Conv, 2FC | 89.53 | -3.85 | - | 90 | 2.91 | - | - | - | - | - |
| LSNN *Bellec et al. (2018)* | 6Conv, 2FC | 92.84 | -3.53 | - | 97.96 | 16.59 | - | - | - | - | - |
| Grad R *Chen et al. (2021)* | 6Conv, 2FC | 92.54 | -0.30 | - | 71.59 | 2.09 | 71.34 | -4.03 | - | 97.65 | 19.45 |
| IMP *Kim et al. (2022b)* | ResNet19 | 93.22 | -0.04 | - | 97.54 | 13.29 | 71.34* | -2.39* | - | 97.54 | 18.69 |
| STDS *Chen et al. (2022a)* | 6Conv, 2FC | 92.49 | -0.35 | - | 88.67 | 5.27 | - | - | - | - | - |
| ESL-SNN *Shen et al. (2023)* | ResNet19 | 91.09 | -1.7 | - | 95 | 2.11 | 73.48 | -0.99 | - | 95 | 14.22 |
| LNP *(ours)* | ResNet19 (Random) | 93.29 ± 0.74 | -2.15 ± 0.19 | 90.48 | 94.32 | 9.36 ± 1.28 | 73.32 ± 0.81 | -3.96 ± 0.39 | 90.48 | 94.32 | 12.04 ± 0.41 |
| | ResNet19 (converted) | | -0.04 ± 0.01 | 93.67 | 98.19 | 22.18 ± 0.2 | | -0.11 ± 0.02 | 94.44 | 98.07 | 31.15 ± 0.28 |
| | HRSNN-STDP | 90.26 ± 0.88 | -0.76 ± 0.07 | 94.03 | 95.06 | 39.01 ± 0.47 | 68.94 ± 0.73 | -2.16 ± 0.28 | 92.02 | 94.21 | 48.21 ± 0.59 |
| | HRSNN-BP | 92.37 ± 0.91 | -0.81 ± 0.08 | | | 35.67 ± 0.41 | 70.12 ± 0.71 | -2.37 ± 0.32 | | | 42.57 ± 0.63 |
| | CHRSNN-STDP | 91.58 ± 0.83 | -0.62 ± 0.07 | 95.04 | 96.68 | 50.37 ± 0.61 | 69.96 ± 0.68 | -1.65 ± 0.21 | 93.47 | 97.04 | 57.44 ± 0.68 |
| | CHRSNN-BP | 93.45 ± 0.87 | -0.74 ± 0.07 | | | 45.32 ± 0.58 | 73.45 ± 0.66 | -1.11 ± 0.23 | | | 50.35 ± 0.64 |

Table 2: Table comparing the performance on the Lorenz 63 and Google datasets. The complete results for the Rossler system and Wind prediction are given in Suppl. Sec. B

| Pruning Method | Model | Training Method | Avg. SOPs (M) | Lorenz63 | | Google Dataset | |
|---|---|---|---|---|---|---|---|
| | | | | RMSE | VPT | RMSE | VPT |
| Unpruned | HRSNN | BP | 815.77 ± 81.51 | 0.248 ± 0.058 | 44.17 ± 6.31 | 0.794 ± 0.096 | 42.18 ± 6.22 |
| | | STDP | 710.76 ± 79.65 | 0.315 ± 0.042 | 35.75 ± 4.65 | 0.905 ± 0.095 | 32.36 ± 3.14 |
| | CHRSNN | BP | 867.42 ± 93.12 | 0.235 ± 0.052 | 47.23 ± 6.02 | 0.782±0.091 | 45.28±5.98 |
| | | STDP | 744.97 ± 80.09 | 0.285 ± 0.021 | 40.17 ± 5.13 | 1.948 ± 0.179 | 19.25 ± 3.54 |
| AP Pruned | HRSNN | BP | 117.52 ± 14.37 | 1.245 ± 0.554 | 31.08 ± 8.23 | 1.457±0.584 | 28.24±6.98 |
| | | STDP | 92.68 ± 10.11 | 1.718 ± 0.195 | 21.10 ± 7.22 | 1.948 ± 0.179 | 19.25 ± 7.59 |
| | CHRSNN | BP | 157.33 ± 18.87 | 1.114 ± 0.051 | 33.97 ± 7.56 | 1.325 ± 0.566 | 26.47±7.42 |
| | | STDP | 118.77 ± 10.59 | 1.596 ± 0.194 | 29.41 ± 7.33 | 1.987 ± 0.191 | 17.68 ± 7.38 |
| LNP Pruned | HRSNN | BP | 22.87 ± 2.27 | 0.691 ± 0.384 | 33.67 ± 6.88 | 0.855±0.112 | 33.14±3.01 |
| | | STDP | 18.22 ± 2.03 | 0.705 ± 0.104 | 32.17 ± 4.62 | 0.917 ± 0.124 | 30.25 ± 3.26 |
| | CHRSNN | BP | 19.14 ± 2.04 | 0.682 ± 0.312 | 39.15 ± 6.27 | 0.832±0.105 | 34.51±2.87 |
| | | STDP | 14.79 ± 1.58 | 0.679 ± 0.098 | 39.24 ± 4.15 | 0.901 ± 0.101 | 32.14± 3.05 |

traditional Artificial Neural Networks (ANNs). The energy consumption of an SNN is calculated as $E = C_E \times$ Total SOPs $= C_E \sum_i s_i c_i$, where $C_E$ represents the energy per SOP, and Total SOPs $= \sum_i s_i c_i$ is the sum of SOPs. Each presynaptic neuron $i$ fires $s_i$ spikes, connecting to $c_i$ synapses, with every spike contributing to one SOP. Again, for sparse SNNs, the energy model is reformulated as Shi et al. (2024): $E = C_E \sum_i \left( s_i \sum_j n_i^{\text{pre}} \wedge \theta_{ij} \wedge n_{ij}^{\text{post}} \right)$, where the set $\left( n_i^{\text{pre}}, \theta_{ij}, n_{ij}^{\text{post}} \right) \in \{0,1\}^3$ indicates the state of the presynaptic neuron, the synapse, and the postsynaptic neuron. A value of 1 denotes an active state, while 0 indicates pruning. The $\wedge$ symbol represents the logical AND operation. Hence, we calculate the "SOP Ratio" between the unpruned and pruned networks as a metric for comparison of the energy efficiency of the pruning methods, which quantifies the energy savings relative to the original, fully connected (dense) network. This ratio provides a meaningful way to gauge the efficiency improvements in sparse SNNs compared to their dense counterparts.

**Readout Layer Pruning:** The read-out layer is task-dependent and uses supervised training. In this paper, we do not explicitly prune the read-out network, but the readout layer is implicitly pruned. The readout layer is a multi-layer (two or three layers) fully connected network. The first layer size is equal to the number of neurons sampled from the recurrent layer. We sample the top 10% of neurons with the greatest betweenness centrality. Thus, as the number of neurons in the recurrent layer decreases, the size of the first layer of the read-out network also decreases. The second layer consists of fixed size with 20 neurons, while the third layer differs between the classification and the prediction tasks such that for classification, the number of neurons in the third layer is equal to the number of classes. On the other hand, the third layer for the prediction task is a single neuron which gives the prediction output.

## 4.2 RESULTS

**Performance Comparison in Classification:** Table 1 shows the comparison of our model with other state-of-the-art pruning algorithms in current literature. It must be noted here that these algorithms are task-dependent and use only synapse pruning, keeping the model architecture fixed. We evaluate the

model on the CIFAR10 & CIFAR100 datasets and observe that our proposed task-agnostic pruning algorithm performs closely to the current state-of-the-art.

**Performance Comparison in Prediction:** The pruned model derived by the LNP method is task-agnostic. As such, we can train the model with either STDP or gradient-based approaches. In this section, we compare the performance of the unsupervised STDP-trained model with the supervised surrogate gradient method to train the pruned HRSNN model (Neftci et al., 2019). First, we plot the evolution of RMSE loss with pruning iterations in Fig. 3(c) when trained and evaluated on the Lorenz 63 dataset. We see that our pruning method shows minimal degradation in performance compared to the baseline activity pruned method. Further on, Table 2 presents the comparative performance of different pruning methods, models, and training methods on the Lorenz 63 and the Google stock prediction datasets. It is clear from the results that the LNP-pruned models generally outperform the Unpruned and AP-pruned counterparts across different models and training methods. Specifically, LNP pruned models consistently exhibit lower SOPs, indicating enhanced computational efficiency while maintaining competitive RMSE and VPT values, which indicate the model's predictive accuracy and validity, respectively. This suggests that employing the LNP pruning method can significantly optimize model performance without compromising the accuracy of predictions.

**Ablation Studies:** We conducted an ablation study, where we systematically examined various combinations of the four sequential steps involved in the LNP method. This study's findings are presented graphically, as illustrated in Fig. 13. At each point (A-E), we train the model and obtain the model's accuracy and average. Synaptic Operations (SOPs) to support ablation studies. The ablation study is done for the HRSNN model, which is trained using STDP and evaluated on the CIFAR10 dataset.

In the figure, different line styles and colors represent distinct aspects of the procedure: the blue line corresponds to Steps 1 and 2 of the LNP process, the orange line to Step 3, and the green line to Step 4. Solid lines depict the progression of the original LNP process $(A \rightarrow B \rightarrow C \rightarrow D \rightarrow E)$, while dotted lines represent the ablation studies conducted at stages B and C. This visual representation enables a clear understanding of the individual impact each step exerts on the model's performance and efficiency. Additionally, it provides insights into potential alternative outcomes that might have arisen from employing different permutations of these steps or omitting certain steps altogether. A detailed description of the ablation study is given in Suppl. Sec. B.5

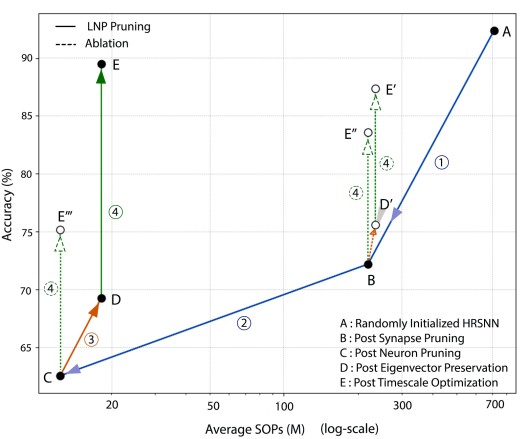

Figure 5: Plot showing Ablation studies of LNP

# 5 CONCLUSION

This research introduced Lyapunov Noise Pruning (LNP), a novel, task-agnostic methodology for designing sparse Recurrent Spiking Neural Networks (RSNNs), emphasizing the balance between computational efficiency and optimal performance. Unlike prevailing methods, LNP starts with a random, densely initialized RSNN model, utilizing the Lyapunov spectrum and spectral graph sparsification methods to prune while maintaining network stability. Experimental results demonstrated that LNP outshone conventional activity-based pruning, reducing computational complexity with fewer neurons and synapses, and maintaining superior accuracy and validity across various datasets, including synthetic and real-world ones. The task-agnostic nature of LNP establishes universally adaptable and robust models without extensive, task-specific adjustments, preserving critical network parameters and optimizing model structures, especially crucial in environments with constrained computational resources. The flatter minima, corresponding to more stable and robust solutions, indicated enhanced stability in the learned dynamics of the model. In summary, LNP represents a significant advancement in neural network design, offering more efficient, stable, and versatile models, suitable for diverse applications and setting the stage for future innovations in the field of neural networks.

## ACKNOWLEDGEMENT

This work is supported by the Army Research Office and was accomplished under Grant Number W911NF-19-1-0447. The views and conclusions contained in this document are those of the authors and should not be interpreted as representing the official policies, either expressed or implied, of the Army Research Office or the U.S. Government.

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

# A  SUPPLEMENTARY SECTION A

## A.1  COMPUTATION OF LYAPUNOV EXPONENTS

We compute LE by adopting the well-established algorithm [57, 58] and follow the implementation in [48, 55]. For a particular task, each batch of input sequences is sampled from a set of fixed-length sequences of the same distribution. We chose this set to be the validation set. For each input sequence in a batch, a matrix $\mathbf{Q}$ is initialized as the identity to represent an orthogonal set of nearby initial states. The hidden states $h_t$ are initialized as zeros.

To track the expansion and the contraction of the vectors of $\mathbf{Q}$, the Jacobian of the hidden states at step t, $\mathbf{J}_t$, is calculated and then applied to the vectors of $\mathbf{Q}$. The Jacobian $\mathbf{J}_t$ can be found by taking the partial derivatives of the RNN hidden states at time $t$, $h_t$, with respect to the hidden states at times $t-1, h_{t-1}$

$$[\mathbf{J}_t]_{ij} = \frac{\partial \mathbf{h}_t^j}{\partial \mathbf{h}_{t-1}^i}.$$

Beyond the hidden states, the Jacobian will depend on the input $x_t$. This dependence allows us to capture the dynamics of a network as it responds to input. The expansion factor of each vector is calculated by updating $\mathbf{Q}$ by computing the $QR$ decomposition at each time step.

$$\mathbf{Q}_{t+1}, \mathbf{R}_{t+1} = QR\left(\mathbf{J}_t \mathbf{Q}_t\right).$$

If $r_t^i$ is the expansion factor of the $i^{th}$ vector at time step $t$ - corresponding to the $i^{\text{th}}$ diagonal element of $\mathbf{R}$ in the QR decomposition- then the $i^{\text{th}}$ LE $\lambda_i$ resulting from an input signal of length $T$ is given by

$$\lambda_k = \frac{1}{T} \sum_{t=1}^{T} \log\left(r_t^k\right)$$

The LE resulting from each input $x^m$ in the batch of input sequences is calculated in parallel and then averaged. For each experiment, the LE was calculated over a fixed number of time steps with $n$ different input sequences. The mean of $n$ resulting LE spectra is reported as the LE spectrum. To normalize the spectra across different network sizes and, consequently the number of LE in the spectrum, we interpolate the spectrum such that it retains the shape of the largest network size.

We follow the principles outlined by Engelken et al. for calculating the Lyapunov spectrum of the discrete-time firing-rate network. Conducting numerical simulations with small perturbations in Recurrent Spiking Neural Networks (RSNNs), particularly in the context of discrete spiking events, requires a specific approach to capture these networks' dynamics accurately.

1. **Network Initialization:** Set up an RSNN with a defined architecture, synaptic weights, and initial neuronal states.

2. **Creating Perturbations**: Generate a slightly perturbed version of the network. This could involve minor adjustments to the initial membrane potentials or other state variables of a subset of neurons.

3. **Simulating Network Dynamics:** Run parallel simulations of the original and the perturbed RSNNs, ensuring they receive identical input stimuli. Since the networks operate on discrete spike events, the state of each neuron is updated based on the inputs it receives and its current potential

4. **Measuring Divergence**: At each time step, measure the difference between the states of the two networks. In the context of spiking neurons, this could involve comparing the spike trains of corresponding neurons in each network. This difference could be quantified using various metrics, such as spike-timing difference, spike count difference, or membrane potential differences.

5. **Tracking the Evolution of the Perturbation**: Observe how the initial small differences evolve. These differences might lead to significantly divergent spiking patterns in a network exhibiting chaotic dynamics.

6. **Estimating Lyapunov Exponents:** Calculate the rate at which the trajectories of the original and perturbed networks diverge. This involves fitting an exponential curve to the divergence data over time. The slope of this curve gives an estimate of the Lyapunov exponent. A positive exponent indicates sensitivity to initial conditions and potential chaotic dynamics. The evolution of the map is given by

$$h_i\left(t_s + \Delta t\right) = f_i = (1 - \Delta t)h_i\left(t_s\right) + \Delta t \sum_{j=1}^{N} J_{ij}\phi\left(h_j\left(t_s\right)\right)$$

In the limit $\Delta t \to 0$, a continuous-time dynamics is recovered. For $\Delta t = 1$, the discrete-time network is obtained. The Jacobian for the discrete-time map is

$$D_{ij}\left(t_s\right) = \left.\frac{\partial f_i}{\partial h_j}\right|_{t=t_s} = (1 - \Delta t)\delta_{ij} + \Delta t \cdot J_{ij}\phi'\left(h_j\left(t_s\right)\right).$$

The full Lyapunov spectrum is again obtained by a reorthonormalization procedure of the Jacobians along a numerical solution of the map.

## A.2 Linearization around Critical Points

Recurrent neural networks (RNNs) are useful tools for learning nonlinear relationships between time-varying inputs and outputs with complex temporal dependencies. Sussillo et al. Sussillo & Barak (2013) have explored the hypothesis that fixed points, both stable and unstable, and the linearized dynamics around them, can reveal crucial aspects of how RNNs implement their computations. Further, they explored the utility of linearization in areas of phase space that are not true fixed points but merely points of very slow movement and presented a simple optimization technique that is applied to trained RNNs to find the fixed and slow points of their dynamics. Linearization around these slow regions can be used to explore, or reverse-engineer, the behavior of the RNN. Similarly, other recent works Sadeh & Rotter (2014) have shown that, for a wide variety of connectivity patterns, a linear theory based on firing rates accurately approximates the outcome of direct numerical simulations of networks of spiking neurons.

Building on these works, we can linearize the HRSNN model around the critical points using the differential equation:

$$\frac{d\boldsymbol{x}}{dt} = -D\boldsymbol{x} + L\boldsymbol{x} + \boldsymbol{b}(t) = A\boldsymbol{x} + \boldsymbol{b}(t) \tag{6}$$

In this equation:

- $\boldsymbol{x}$ represents the vector of firing rates of neurons in the network. - $D$ is a diagonal matrix indicating neurons' intrinsic leak or excitability. - $L$ is the Lyapunov matrix. - $\boldsymbol{b}(t)$ denotes external inputs, including biases. - $A$ is a matrix defined as $A = -D + L$.

The spike frequency of untrained SNNs is used to approximate the firing rates ($\boldsymbol{x}$) in the network. In an untrained SNN, the firing rates can be considered as raw or initial responses to inputs before any learning or adaptation has occurred. This approximation is useful for constructing a linearized model as it provides a baseline from which the effects of learning, pruning, and other dynamics can be analyzed. Each diagonal element $D_{ii}$ of the matrix $D$ quantifies how the firing rate of neuron $i$ changes over time without external input or interaction. These elements can be determined based on the inherent properties of the neurons in the network, such as their leakiness or excitability. The Lyapunov matrix $L$ encapsulates the impact of neighboring nodes on each edge regarding their Lyapunov exponents. The elements of $L$ are calculated using the harmonic mean of the Lyapunov exponents of the neighbors of nodes $i$ and $j$ as detailed in your method. This matrix represents how the dynamics of one neuron affect its neighbors, influenced by the network's overall stability and dynamical behavior. In summary, the linearized model provided by equation 6 is a simplification that helps to understand the fundamental dynamics of the SNN. It uses the initial, untrained spike frequencies to establish a baseline for the network's behavior, and the matrices $D$ and $L$ are calculated based on the intrinsic properties of the neurons and their interactions, respectively.

We aim to create a sparse network ($A^{\text{sparse}}$) with fewer edges while maintaining dynamics similar to the original network. The sparse network is thus represented as:

$$\frac{d\boldsymbol{x}}{dt} = A^{\text{sparse}}\boldsymbol{x} + \boldsymbol{b}(t) \quad \text{such that} \quad \left|\boldsymbol{x}^T\left(A^{\text{sparse}} - A\right)\boldsymbol{x}\right| \le \epsilon\left|\boldsymbol{x}^T A\boldsymbol{x}\right| \quad \forall \boldsymbol{x} \in \mathbb{R}^N \tag{7}$$

for some small $\epsilon > 0$. When the network in Eq. 6 is driven by independent noise at each node, we define $\boldsymbol{b}(t) = \boldsymbol{b} + \sigma\boldsymbol{\xi}(t)$, where $\boldsymbol{b}$ is a constant input vector, $\boldsymbol{\xi}$ is a vector of IID Gaussian white noise, and $\sigma$ is the noise standard deviation. Let $\Sigma$ be the covariance matrix of the firing rates in response to this input. The probability $p_{ij}$ for the synapse from neuron $j$ to neuron $i$ with the Lyapunov exponent $l_{ij}$ is defined as:

$$p_{ij} = \begin{cases} \rho l_{ij}(\Sigma ii + \Sigma jj - 2\Sigma ij) & \text{for } w_{ij} > 0 \text{ (excitatory)} \\ \rho|l_{ij}|(\Sigma ii + \Sigma jj + 2\Sigma ij) & \text{for } w_{ij} < 0 \text{ (inhibitory)} \end{cases} \tag{8}$$

Here, $\rho$ determines the density of the pruned network. The pruning process independently preserves each edge with probability $p_{ij}$, yielding $A^{\text{sparse}}$, where $A_{ij}^{\text{sparse}} = A_{ij}/p_{ij}$, with probability $p_{ij}$ and 0 otherwise. For the diagonal elements, denoted as $A_{ii}^{\text{sparse}}$, representing leak/excitability, we either retain the original diagonal, setting $A_{ii}^{\text{sparse}} = A_{ii}$, or we introduce a perturbation, $\Delta_i$, defined as the difference in total input to neuron $i$, and adjust the diagonal as $A_{ii}^{\text{sparse}} = A_{ii} - \Delta_i$. Specifically, $\Delta_i = \sum_{j\neq i}|A_{ij}^{\text{sparse}}| - \sum_{j\neq i}|A_{ij}|$. This perturbation, $\Delta_i$, is typically minimal with a zero mean and is interpreted biologically as a modification in the excitability of neuron $i$ due to alterations in total input, aligning with the known homeostatic regulation of excitability.

## A.3 BASELINE PRUNING METHODS

Activity pruning is a technique employed to optimize neural network models by iteratively removing the least active neurons. In this approach, outlined as the Iterative Activity Pruning algorithm, the process starts with an initial Recurrent Spiking Neural Network (RSNN) model $M$. The algorithm operates by first evaluating each neuron's activity level, followed by pruning a certain percentage (determined by the pruning rate $r$) of neurons that exhibit the lowest activity. Post pruning, the model $M$ is retrained to compensate for the loss of neurons, forming an updated model $M'$. This cycle of pruning and retraining continues until either a maximum number of iterations $T$ is reached, or the performance of the pruned model drops below 10% of the original, unpruned model's performance. The goal of this method is to refine the model by removing less critical neurons while maintaining or enhancing overall performance. This technique is validated by comparing its efficacy in classifying datasets like CIFAR10 & CIFAR100 against other state-of-the-art pruning algorithms. The detailed algorithm is given in Algorithm 4. We also evaluate our model for classifying the CIFAR10 & CIFAR100 datasets and compare the results with current task-dependent state-of-the-art pruning algorithms. The results are shown in Table 1.

## A.4 DATASETS

### A.4.1 LORENZ SYSTEM

The Lorenz system is a non-linear, three-dimensional system that can be described as follows:

$$dx/dt = \sigma(y - x)$$
$$dy/dt = x(\rho - z) - y$$
$$dz/dt = xy - \beta z$$

when $\sigma = 10$, $\beta = 8/3$, and $\rho = 28$, the system has chaotic solutions. The experimental setup, same as Xu et al. (2018a), was used in this paper, and the fourth-order Runge-Kutta method was used to generate samples. Table 3 summarizes the details of the experimental setup.

Table 3: Details of the experimental setup for the Lorenz system.

| Parameter | Value |
|---|---|
| Number of samples | 20000 |
| Initial state | $[12, 2, 9]$ |
| Step size | 0.01 |
| Number of training samples | 11250 |
| Number of validation samples | 3750 |
| Number of test samples | 5000 |

Table 4: Details of the experimental setup for the Rossler system.

| Parameter | Value |
|---|---|
| Number of samples | 12700 |
| Initial state | $[1, 1, 1]$ |
| Step size | 0.03 |
| Number of discarded samples | 7700 |
| Number of training samples | 3000 |
| Number of validation samples | 1000 |
| Number of test samples | 1000 |

### A.4.2   ROSSLER SYSTEM

The Rossler system is a classical system, consisting of three nonlinear ordinary differential equations and can be defined by:

$$dx/dt = -y - z$$
$$dy/dt = x + ay$$
$$dz/dt = b + z(x - c)$$

when $a = 0.15, b = 0.2$, and $c = 10$, the system shows chaotic behavior. To compare the performance of MFRFNN with other methods under the same condition, the experimental setup, same as Xu et al. (2018b), was used for the Rossler system. In this setup, the fourth-order Runge-Kutta method was employed for sample generation. Some of the samples were discarded to eliminate the transient influence of the initial condition. Table 5 presents the details of the experimental setup for the Rossler system.

### A.4.3   GOOGLE STOCK PRICE PREDICTION PROBLEM

Stock price prediction is a non-linear and highly volatile problem. In this problem, the future value of Google stock price is predicted using the current price as defined by.

$$\hat{y}(t) = f(y(t - 1))$$

The dataset was obtained from Yahoo Finance during six years from 19-August-2004 to 21-September-2010 as in Samanta et al. (2020). The training set consisted of 1529 samples, and the test set of 900 samples. To evaluate the performance of MFRFNN on another real-world time series, we compared its performance with the same RFNNs and FNNs used in Box-Jenkins and wind speed prediction datasets.

### A.4.4   WIND PREDICTION

The wind speed prediction problem is a non-linear, dynamic, and volatile problem in which the future value of wind speed is predicted using the current wind speed and wind direction. The dataset is obtained from the Iowa Department of Transport's website.1 The data was collected from the Washington station during a one-month period (February 2011), sampled every ten minutes, and averaged hourly. There are 500 samples in the training set and 1000 samples in the test set Samanta et al. (2020). This dataset is more challenging than the Box–Jenkins dataset due to the existence of

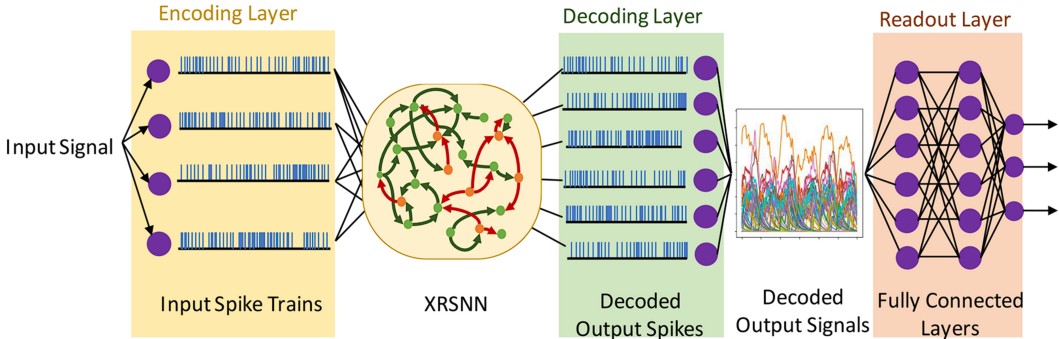

Figure 6: Block Diagram showing the methodology using HRSNN for prediction

noise. This experiment compared the proposed method's performance with the same algorithms we used in the Box–Jenkins dataset.

## A.5 Calculation of SOPs

In evaluating the energy efficiency of neuromorphic chips, a key metric is the average energy consumption for transmitting a single spike across a synapse, as highlighted by some recent works Furber (2016),Shi et al. (2024). This metric is particularly important due to the substantial energy expenditure involved in synapse processing, which significantly impacts the overall energy usage. For a theoretical analysis that is independent of specific hardware, we consider the average energy for a single spike-synapse transmission as a fixed constant. The total energy consumption of a Spiking Neural Network (SNN) model can be approximated by tallying the synaptic operations (SOPs) required, analogous to counting floating-point operations (FLOPs) in Artificial Neural Networks (ANNs). Our model for calculating the SNN's energy consumption is expressed as:

$$E = C_E \cdot \text{SOP} = C_E \sum_i s_i c_i \tag{9}$$

Here, $C_E$ represents the energy usage per SOP, and $\text{SOP} = \sum_i s_i c_i$ is the cumulative count of synaptic operations. In any given presynaptic neuron $i$, $s_i$ signifies the spike count emitted by that neuron, while $c_i$ indicates its synaptic connections. Each spike transmission from this neuron triggers a synaptic operation as it reaches the postsynaptic neurons, contributing to the energy expenditure.

In the context of sparse SNNs, the energy consumption model is modified as follows:

$$E = C_E \sum_i \left( s_i \sum_j n_i^{\text{pre}} \wedge \theta_{ij} \wedge n_{ij}^{\text{post}} \right) \tag{10}$$

In this equation, for every synaptic link from the $i$-th presynaptic neuron to its $j$-th postsynaptic neuron, the tuple $\left( n_i^{\text{pre}}, \theta_{ij}, n_{ij}^{\text{post}} \right)$, each element of which can be either 0 or 1, represents the states of the presynaptic neuron, the synapse, and the postsynaptic neuron, respectively, here, 1 indicates an active state, and 0 denotes a pruned state, the symbol $\wedge$ stands for the logical AND operation.

## A.6 HRSNN Model

**HRSNN Model Architecture:** Fig. 6 shows the overall architecture of the prediction model. Using a rate-encoding methodology, the time-series data is encoded to a series of spike trains. This high-dimensional spike train acts as the input to HRSNN. The output spike trains from HRSNN act as the input to a decoder and a readout layer that finally gives the prediction results. For the classification task, we use a similar method. However, we do not use the decoding layer for the signal but directly feed the output spike signals from HRSNN into the fully connected layer.

**Readout Layer:** The read-out layer is task-dependent and uses supervised training. In this paper, we do not explicitly prune the read-out network, but the readout layer is implicitly pruned. The readout layer is a multi-layer (two or three layers) fully connected network. The first layer size is equal to the

Table 5: Table showing the relative performance change when including the readout layer in the calculations

|  | Neuron Sparsity Change | Synapse Sparsity Change | Avg SOP change CIFAR10 | Avg SOP change CIFAR100 |
|---|---|---|---|---|
| HRSNN | -0.13 | -0.18 | -0.09 | -0.14 |
| CHRSNN | -0.08 | -0.13 | -0.1 | -0.16 |

number of neurons sampled from the recurrent layer. We sample the top 10% of neurons with the greatest betweenness centrality. Thus, as the number of neurons in the recurrent layer decreases, the size of the first layer of the read-out network also decreases. The second layer consists of fixed size with 20 neurons, while the third layer differs between the classification and the prediction tasks such that for classification, the number of neurons in the third layer is equal to the number of classes. On the other hand, the third layer for the prediction task is a single neuron which gives the prediction output. The table 5 shows the relative change in sparsity when including/excluding the readout layer for calculations:

## A.7 DYNAMIC CHARACTERIZATION USING LYAPUNOV SPECTRA

To show the principal dynamic characteristic of the LNP-HRSNN model, we plot the full Lyapunov spectrum of the HRSNN model for three different cases - the unpruned network, the pruned network, and the trained pruned network. We refer to the methodologies discussed in recent works Vogt et al. (2020); Engelken et al. (2023). The Lyapunov Spectrum provides valuable additional insights into the collective dynamics of firing-rate networks. We plot the evolution of Lyapunov spectra with different initialization parameters. We plotted the Lyapunov spectrum, with three different probabilities of synaptic connection for the initial network (p=0.001, p=0.01, p=0.1). We plot the Lyapunov exponents ($\lambda_i$) vs the normalized indices $i/N$ described as follows:

- $\lambda_i$: This axis represents the Lyapunov exponents. A positive exponent indicates chaos, meaning that two nearby trajectories in the phase space will diverge exponentially. A negative exponent suggests that trajectories converge, and zero would imply neutral stability.

- **i/N**: This axis is likely indexing the normalized Lyapunov exponents, with $i$ being the index of a particular exponent and $N$ being the total number of exponents calculated. For a system with $N$ dimensions, there are $N$ Lyapunov exponents.

The graph shows three plots of the Lyapunov spectrum for different stages of pruning and training:

1. **Unpruned**: This plot represents the Lyapunov spectrum of the neural network before any pruning has been done. The spectrum shows a range of Lyapunov exponents from positive to negative values, indicating that the network likely has both stable and chaotic behaviors. More notably, the more sparse initialized models (p = 0.1) were more unstable as the largest Lyapunov exponent is positive.

2. **Pruned Pre-Training**: This plot shows the Lyapunov spectrum after the network has been pruned but before it has been trained again. Pruning is a process in which less important connections (weights) in a neural network are removed, which can simplify the network and potentially lead to more efficient operation without significantly impacting performance.

3. **Pruned Post-Training**: This plot illustrates the Lyapunov spectrum after the network has been pruned and then trained again. Retraining the network after pruning allows it to adjust the remaining weights to compensate for the removed connections, which can restore or even improve performance.

We can infer the following from the plots: **Impact of Pruning:** Comparing the 'Unpruned' to the 'Pruned Pre-Training' plot, it seems that pruning increases the stability of the system as the Lyapunov exponents become more negative (less chaotic). This might suggest that pruning reduces the complexity of the dynamics.

**Training Effect:** The 'Pruned Post-Training' plot shows that after re-training, the exponents are less negative compared to 'Pruned Pre-Training', which may indicate that the network is able to regain some dynamical complexity or expressive power through training.

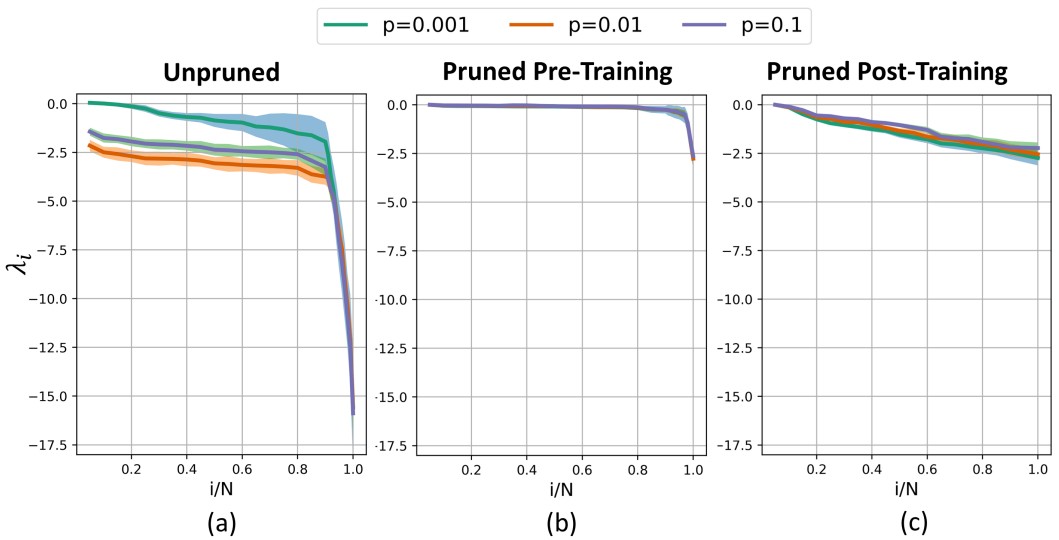

Figure 7: Lyapunov Spectrum of the HRSNN model in three stages of pruning and training and different levels of sparsity for initialization: (a) the Lyapunov Spectrum of unpruned HRSNN model with a probability of connection $p = 0.1, 0.01, 0.001$ (b) the Lyapunov Spectrum after LNP pruning (c) the Lyapunov Spectrum after training of the pruned model

**Initialization Synapse Density:** Different initialization synapse densities (arising from different p) have different effects on the spectrum. High $p$) tends to result in more negative exponents, suggesting greater stability but potentially less capacity to model complex patterns.

**Network Robustness:** The fact that the Lyapunov spectrum does not dramatically change in shape across the three stages, especially in the 'Pruned Post-Training' stage, might imply that the network is robust to pruning, retaining its general dynamical behavior while likely improving its efficiency.

**Convergence of Spectra:** The convergence of the spectra for different values of $p$ after pruning, both pre-and post-training, suggests that regardless of the initial density of connections, the network may evolve towards a structure that is similarly efficient for the task it is being trained on. This indicates that the LNP algorithm can efficiently prune a network irrespective of the initial density and also make the final model more stable.

Overall, the Lyapunov Spectrum serves as useful tool for understanding how network pruning and re-training affect the dynamics of neural networks, which is important for optimizing neural networks for better performance and efficiency.

**Comparison with Randomly Sparse Initialization**: We can get an idea of the principal dynamic characteristic of LNP-HRSNN from the Lyapunov spectrum of the randomly initialized network with a very low probability of connection (p=0.001). We see that such networks were more unstable as the largest Lyapunov exponent was positive. This unstable behavior might be because randomly generated networks lack structured connections that might otherwise guide or constrain the flow of neural activity. Without these structures, the network is more likely to exhibit erratic behavior as the activity patterns are less predictable and more susceptible to amplification of small perturbations, making them more unstable.

## B SUPPLEMENTARY SECTION B

### B.1 ADDITIONAL RESULTS

**Performance Evaluation**

For a more extensive performance evaluation of the LNP pruning method, we test the model on 4 different datasets and note the RMSE loss and the VPT. The results are shown in Table 6. From the

Table 6: Performance of HRSNN and CHRSNN models using different pruning methods. Each model is trained on the first 200 timesteps and predicts the next 100 timesteps.

| Pruning Method | Model | SOPs | Synthetic | | | | Real-world | | | |
| --- | --- | --- | --- | --- | --- | --- | --- | --- | --- | --- |
| | | | Lorenz63 | | Rossler | | Google Stock | | Wind | |
| | | | RMSE | VPT | RMSE | VPT | RMSE | VPT | RMSE | VPT |
| Unpruned | *HRSNN* | $710.76 \pm 79.65$ | $0.315 \pm 0.042$ | $35.75 \pm 4.65$ | $0.142 \pm 0.019$ | $41.32 \pm 5.32$ | $0.905 \pm 0.095$ | $32.36 \pm 3.14$ | $1.098 \pm 0.033$ | $28.36 \pm 3.35$ |
| | *CHRSNN* | $744.97 \pm 80.09$ | $0.285 \pm 0.021$ | $40.17 \pm 5.13$ | $0.125 \pm 0.017$ | $44.17 \pm 4.95$ | $1.098 \pm 0.091$ | $36.58 \pm 3.89$ | $1.181 \pm 0.29$ | $27.95 \pm 3.02$ |
| AP Pruned | *HRSNN* | $92.68 \pm 10.11$ | $1.718 \pm 0.195$ | $21.10 \pm 7.22$ | $0.989 \pm 0.127$ | $29.55 \pm 6.95$ | $1.948 \pm 0.179$ | $19.25 \pm 3.54$ | $2.146 \pm 0.081$ | $14.25 \pm 3.74$ |
| | *CHRSNN* | $118.77 \pm 10.59$ | $1.596 \pm 0.194$ | $29.41 \pm 7.33$ | $0.879 \pm 0.131$ | $37.25 \pm 6.42$ | $1.987 \pm 0.191$ | $17.68 \pm 2.59$ | $2.228 \pm 0.075$ | $16.98 \pm 3.22$ |
| LNP Pruned | *HRSNN* | $18.22 \pm 2.03$ | $0.705 \pm 0.104$ | $32.17 \pm 4.62$ | $0.368 \pm 0.051$ | $40.15 \pm 5.12$ | $0.917 \pm 0.124$ | $30.25 \pm 3.26$ | $0.314 \pm 0.049$ | $27.98 \pm 1.28$ |
| | *CHRSNN* | $14.79 \pm 1.58$ | $0.679 \pm 0.098$ | $39.24 \pm 4.15$ | $0.314 \pm 0.042$ | $47.36 \pm 4.89$ | $0.901 \pm 0.101$ | $32.14 \pm 3.05$ | $0.301 \pm 0.035$ | $28.06 \pm 2.25$ |

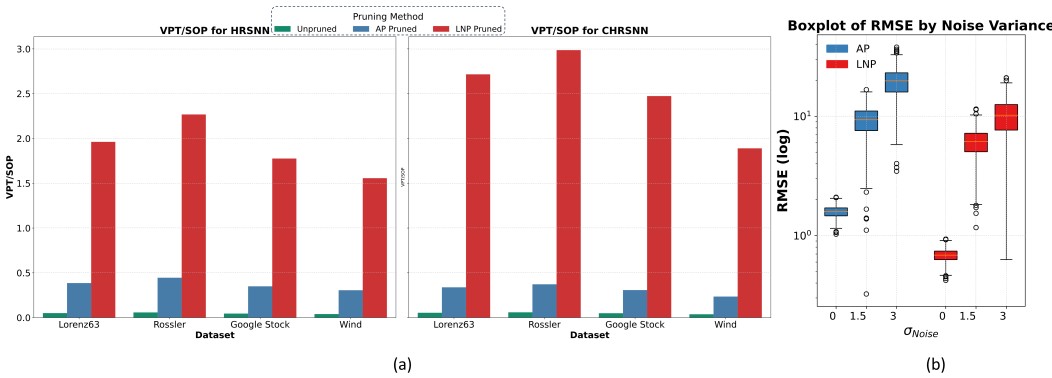

Figure 8: (a) Bar graph showing the comparative analysis of how efficiency (VPT/SOPs) changes for CHRSNN and HRSNN models for the different pruning methods (b) Box plot showing the change in performance of the three trained models obtained by the three pruning methods - AP, and LNP

table, we can see that models undergoing LNP Pruning generally outperform their counterparts across a variety of datasets and metrics. For instance, focusing on the Real-world dataset 'Wind' under the RMSE metric, the LNP Pruned models, both HRSNN and CHRSNN, exhibit the lowest error rates with $0.314 \pm 0.049$ and $0.301 \pm 0.035$, respectively. The underlined values across LNP Pruned models indicate a consistently superior performance, representing the lowest RMSE and VPT values among the pruned and unpruned models across all datasets. Additionally, despite having a smaller FLOPS value, indicative of a lighter model, LNP pruned models, particularly CHRSNN with only 0.018 FLOPS, achieve superior or comparable performance against the unpruned models, demonstrating the efficacy of LNP in maintaining model accuracy while reducing computational complexity. The marked efficiency and performance enhancement accentuate the viability of LNP as a potent pruning strategy, rendering it optimal for environments where computational resources are a constraint.

**Comparing Efficiency** We analyzed the performance of different pruning methods: Unpruned, Activity Pruned, and LNP Pruned, on four datasets: Lorenz63, Rossler, Google Stock, and Wind, using HRSNN and CHRSNN models, as shown in Suppl. Fig 8. The bar graphs illustrate that the LNP Pruned method performs the best in terms of efficiency (defined as the ratio of VPT to FLOPS) across all datasets and models, highlighting its effectiveness in improving computational efficiency and resource use. The LNP Pruned method is particularly distinguished in CHRSNN models, where it achieves high VPT/FLOPS values in the Rossler and Lorenz63 datasets. The efficacy of this method stems from its capability to retain critical network parameters while discarding the redundant ones, thereby ensuring optimized model performance without compromising accuracy. Such optimization is imperative in practical scenarios where computational resources are constrained, necessitating the development of efficient and effective models.

**Comparison of Performance with Noise**

Next, we test the stability of the models with varying levels of input noise. The results are plotted in Fig. 9 offers an insightful depiction of the Prediction RMSE, represented on a logarithmic scale, across varying input Signal-to-Noise Ratios (SNRs) for both HRSNN and CHRSNN models, subjected to different pruning methods: Unpruned, AP Pruned, and LNP Pruned. The key observation

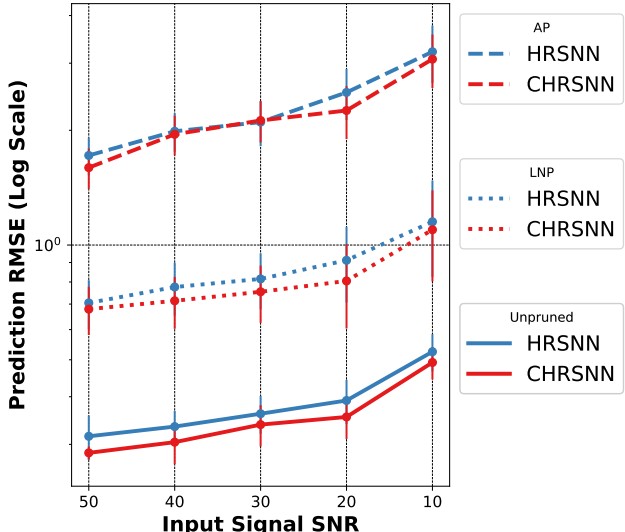

Figure 9: Performance Analysis of Pruned and Unpruned Models with Varied Input SNR Levels

Table 7: Performance of the pruned and unpruned models with different input noise levels.

| | | SNR (dB) | | | | |
|---|---|---|---|---|---|---|
| | | 50 | 40 | 30 | 20 | 10 |
| Unpruned | HRSNN | 0.315±0.042 | 0.334±0.033 | 0.361±0.041 | 0.391±0.052 | 0.425±0.059 |
| | CHRSNN | 0.285±0.011 | 0.304±0.038 | 0.338±0.042 | 0.354±0.044 | 0.362±0.048 |
| AP | HRSNN | 1.718±0.195 | 1.988±0.229 | 2.101±0.284 | 2.514±0.386 | 3.114±0.551 |
| | CHRSNN | 1.596±0.194 | 1.952±0.233 | 2.122±0.264 | 2.254±0.357 | 2.974±0.492 |
| LNP | HRSNN | 0.705±0.104 | 0.776±0.123 | 0.815±0.136 | 0.952±0.205 | 1.421±0.326 |
| | CHRSNN | 0.679±0.098 | 0.714±0.111 | 0.754±0.129 | 0.825±0.201 | 1.397±0.294 |

from the figure is the universal increase in RMSE with the decrease in SNR, reflecting the inherent challenges associated with noise in input signals. Among the evaluated methods, AP Pruned models exhibit the highest RMSE across all SNR levels, indicating suboptimal performance under noise. In contrast, Unpruned models maintain the lowest RMSE, particularly at higher SNRs, showcasing their robustness to noise. The LNP Pruned method achieves a balanced performance, demonstrating its capability to maintain considerable accuracy under noisy conditions while optimizing computational efficiency, hence affirming its practical applicability where a balance between accuracy and efficiency is essential. The visualization succinctly encapsulates the comparative performance dynamics, providing valuable insights into the interdependencies between noise levels, pruning methods, and model types.

**Final Timescales of the Pruned Network** For this paper, all the experiments are initiated with a 5000-neuron network. We have two different starting architectures - the HRSNN model or the CHRSNN model. As discussed in the text, the LNP Pruning algorithm, not only prunes the network but also optimizes the timescales. The final resultant timescale of this pruning method for a given initialization of the model is shown in Fig. 10.

## B.2 USING THE LNP ALGORITHM ON FEEDFORWARD NETWORKS

**Modified Algorithm:** In this section, we extend the Lyapunov Noise Pruning Method for feedforward Networks. We note that this is not a trivial task, as the original method was engineered to exploit the recurrence present in the neuronal dynamics. Hence, for the case of feedforward neural networks, we

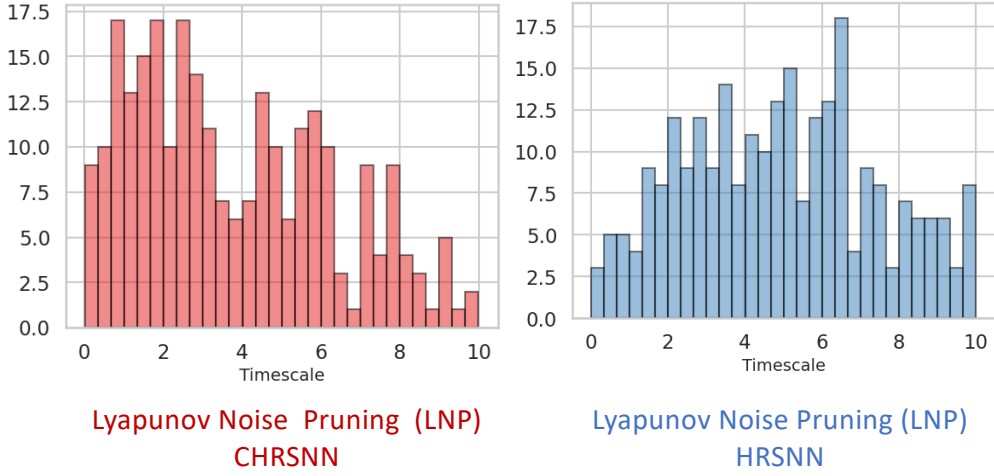

Figure 10: Figure showing the final output timescales of the LNP pruning algorithm on the HRSNN and CHRSNN networks

solely focus on ResNet-based architectures, i.e., models with added skip connections. Thus, the four steps for the LNP-based pruning algorithm are modified as follows:

- **Step 1: Unstructured Pruning of Synapses:** The process of synapse pruning is the same, where the synapses are pruned using the noise-based pruning algorithm described earlier.

- **Step 2: Structured Pruning of Channels:** For feedforward SNNs like ResNets, we use channels instead of neurons for structured pruning. Let us denote the $k$-th channel of the $l$-th layer in the network as $x_k^l$. In each iteration, the average membrane potential of this channel on the training set is computed as

$$x_k^l = \frac{1}{T}(\sum_{t=1}^{T} \|V_k^l(t)\|)$$

Here, $T$ is the timestep of the SNN, and $|V_k^l(t)|$ represents the L1 norm of the membrane potential of the channel's feature map at time $t$. The feature map's membrane potential for a specific neuron at time $t$ is denoted as $V_t^{ij}$, with the channel's membrane potential being $|V_k^l(t)| = [V_t^{ij}]_{h \times w}$. A positive $V_t^{ij}$ indicates an excitatory postsynaptic potential (EPSP), whereas a negative value signifies an inhibitory postsynaptic potential (IPSP).

To optimize the network, we prune convolutional kernels based on the calculated channel importance scores. Channels with lower scores are deemed redundant and are more likely to be pruned. This method ensures a balanced reduction of redundant components across the network, enhancing compression and maintaining performance. We use a mask to track the structure of the network after pruning, where a value of 0 indicates a pruned channel and 1 signifies a retained channel.

- **Step 3: Norm Preservation using Skip Connections:** Norm preservation in ResNets is the property that the norm of the gradient of the network with respect to the input is close to the norm of the gradient with respect to the output. This property is desirable because it means that the network can avoid the vanishing or exploding gradient problem, which can hamper the optimization process. Norm preservation is facilitated by the skip connections in the residual blocks, which allow the gradient to flow directly from the output to the input. Norm preservation is also enhanced as the network becomes deeper because the residual blocks act as identity mappings that preserve the norm of the gradient. After the unstructured and structured pruning in Steps 1 and 2, we reintroduce some additional skip connections for norm preservation.

Table 8: Table showing the performance of LNP on Feedforward Neural Networks

| Model | CIFAR10 | | | | CIFAR100 | | | |
|---|---|---|---|---|---|---|---|---|
| | Baseline Accuracy | Acuracy | Neuron Sparsity | Synapse Sparsity | Baseline Accuracy | Acuracy | Neuron Sparsity | Synapse Sparsity |
| ResNet19 (untrained) | 92.11± 0.9 | -2.15± 0.19 | 90.48 | 94.32 | 73.32± 0.81 | -3.56± 0.39 | 90.48 | 94.32 |
| ResNet19* (converted) | 93.29± 0.74 | -0.04± 0.01 | 93.67 | 98.19 | 74.66± 0.65 | -0.11± 0.02 | 94.44 | 98.07 |

- **Step 4: Neuronal Timescale Optimization:** Similar to the previous case, we optimize the hyperparameters of the spiking ResNet using the Lyapunov spectrum-based Bayesian Optimization technique.

**Experiments and Evaluations:** Now, since the model pruning heavily relies on the skip connections for the norm preservation step, hence, for evaluating the performance of the LNP on the feedforward Neural networks, we look into the ResNet-based model. We evaluate the model for CIFAR10 and CIFAR100 datasets with two different initializations: For the first case, we use the untrained ResNet19 network, then prune it and finally train it and get the model performance. For the second case, we use the pre-trained SNN, which is converted from an ANN using ANN-to-SNN conversion techniques, and then use the LNP pruning on that ResNet model. The results are shown in Table 8. We see that LNP gives extremely good results when used on the SNN, which is converted from the ANN model. Hence opening up another possible application of the LNP pruning method.

## B.3 GENERALIZATION PROPERTIES

As discussed earlier, our proposed LNP pruning algorithm is task-agnostic and does not use a dataset to train while pruning. This pre-training pruning algorithm makes the model extremely generalizable as opposed to current state-of-the-art pruning methods, which require iterative pruning and retraining. This iterative process makes these pruned models extremely overfitted to the dataset it is trained on, and would thus require retraining and repruning of the model for each dataset.

Our LNP algorithm starts with an untrained dense network, and the pruning process does not consider task performance while removing network connections (and neurons) and neuronal time scales. Consequently, the end pruned network does not overfit to any particular dataset; rather generalizable to many different datasets, and even different tasks. On the other hand, prior SNN pruning papers, start with a pre-trained DNN model on a given task which is converted to an SNN. During pruning, the network is simultaneously pruned and re-trained with the dataset (at each pruning step) to minimize performance loss on that dataset. This makes the model overfitted (and hence, shows good results) to that dataset. We use the following terminology for the different pruning cases:

- *Untrained SNN:* The SNN model is not trained before pruning, and the parameters are randomly initialized. Only the final pruned network is then trained

- *Converted SNN*: A standard DNN was trained on a dataset and then converted to an SNN with the same architecture using DNN-to-SNN conversion methods.

To verify our conjecture, we empirically study the following questions:

1. **How does our pruning method perform on a converted SNN model like prior works?**
   - **Our Approach 1 (ResNet converted):** We consider two cases. First, we use our pruning method on a dense Spiking ResNet, which is converted from a DNN ResNet (trained on the CIFAR10 and CIFAR100 datasets). We continue with our approach where the pruning iterations do not consider task performance. We observe that when starting from a pre-trained dense model like prior works, the performance and sparsity of the network pruned with LNP are better than prior pruning methods.
   - **Our Approach 2 (ResNet untrained):** We start with an untrained ResNet with randomly initialized weights and parameters. They use our LNP algorithm without training the model at any iteration of the pruning process. The final pruned network

Table 9: Table showing the generalization performance of the LNP algorithm compared to other baseline pruning methods

| Pruning Method | Architecture | Neuron Sparsity | Synapse Sparsity | CIFAR10 Accuracy | CIFAR100 Accuracy |
|---|---|---|---|---|---|
| *IMP* *Kim et al. (2022b)* | *ResNet19 (iterative pruning and retraining)* | - | 97.54 | 93.18 | 68.95 |
| *LNP (Ours)* | *ResNet19 (untrained)* | 90.48 | 94.32 | 89.96 ± 0.71 | 69.76 ± 0.42 |
| *LNP (Ours)* | *ResNet (pre-trained)* | 93.67 | 98.19 | 93.25 ± 0.73 | 74.55 ± 0.63 |

with optimized hyperparameters (optimized using the Lyapunov spectrum) is then trained and tested on the CIFAR10 and CIFAR100 datasets

- **Approach by IMP Kim et al. (2022b):** Iterative Magnitude Pruning (IMP) is a technique for neural network compression that involves alternating between pruning and retraining steps. In this process, a small percentage of the least significant weights are pruned, and the network is then fine-tuned to recover performance. This iterative cycle continues until a desired level of sparsity is reached, potentially enhancing network efficiency with minimal impact on accuracy.

2. **How do other task-dependent pruning methods perform when the network pruned for one task is used for a different task?**

   - To compare this generalizability property, we took the pruned model from the works of Kim et al. (2022b), which is optimized for CIFAR10. We next train their pruned model on CIFAR100 sparsity, accuracy and SOPs. We compare the performance with the performance of our pruned untrained ResNet model (with random weights) and pruned converted ResNet (as described earlier). We see that the untrained ResNet generated by our LNP-based pruning can generalize better to the CIFAR100 dataset and show better performance. Moreover, we see that using the converted ResNet, we see better performance and efficiency of the pruned ResNet (converted) than the IMP-based pruning model on both CIFAR10 and CIFAR100 datasets.

The results are shown in the Table 9.

## B.4 SCATTER PLOTS

To get a better understanding of the performance vs efficiency of the final pruned models of the LNP method compared to the state-of-the-art pruning models. We plot the accuracy vs average SOPs for the CIFAR10, and CIFAR100 classification tasks and the VPT vs. average. SOPs for the Lorenz63 prediction task. The results are shown in Fig. 11 and 12. We observe that the LNP-based methods always show better performance with lower SOPs compared to other state-of-the-art pruning methods.

## B.5 ABLATION STUDY

We conducted an ablation study, where we systematically examined various combinations of the four sequential steps involved in the LNP method. This study's findings are presented graphically, as illustrated in Fig. 13. At each point (A-E), we train the model and obtain the model's accuracy and average. Synaptic Operations (SOPs) to support ablation studies. The ablation study is done for the HRSNN model, which is trained using STDP and evaluated on the CIFAR10 dataset.

In the figure, different line styles and colors represent distinct aspects of the procedure: the blue line corresponds to Steps 1 and 2 of the LNP process, the orange line to Step 3, and the green line to Step 4. Solid lines depict the progression of the original LNP process ($A \rightarrow B \rightarrow C \rightarrow D \rightarrow E$), while dotted lines represent the ablation studies conducted at stages B and C. This visual representation enables a clear understanding of the individual impact each step exerts on the model's performance and efficiency. Additionally, it provides insights into potential alternative outcomes that might have arisen from employing different permutations of these steps or omitting certain steps altogether.

Below is a detailed discussion of the figure:

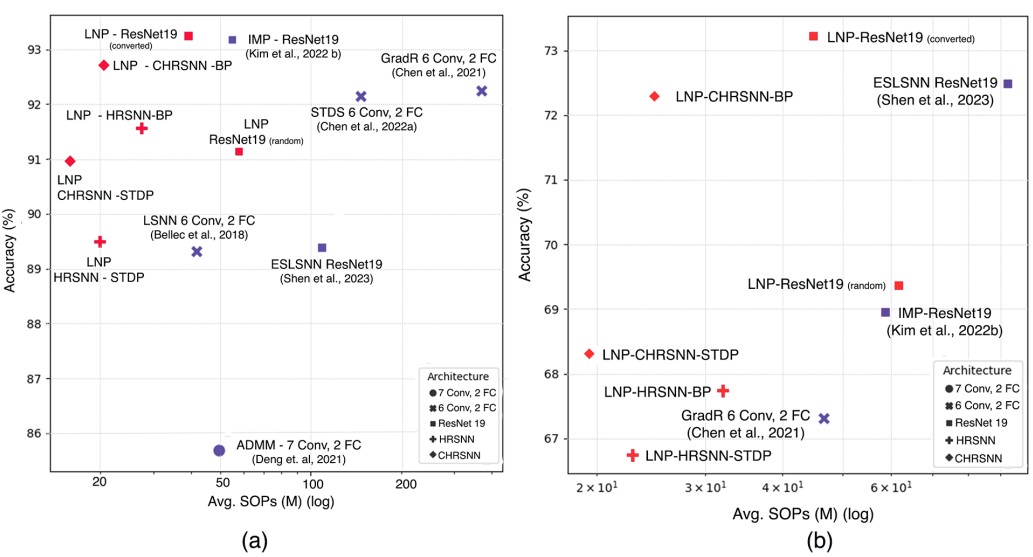

Figure 11: Scatter Plot showing the Performance vs SOPs for (a) CIFAR10 and (b) CIFAR100

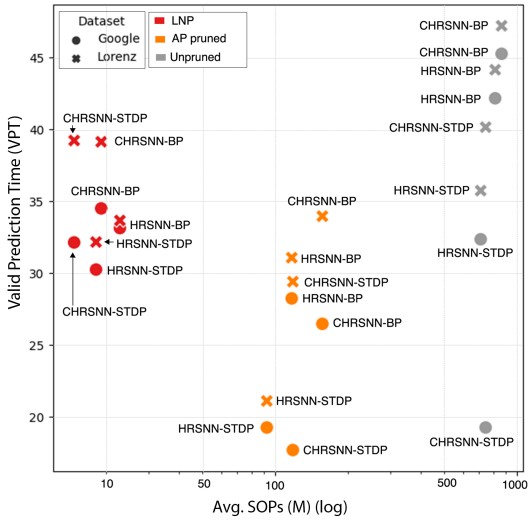

Figure 12: Scatter Plot showing the Performance vs SOPs for Lorenz63 dataset

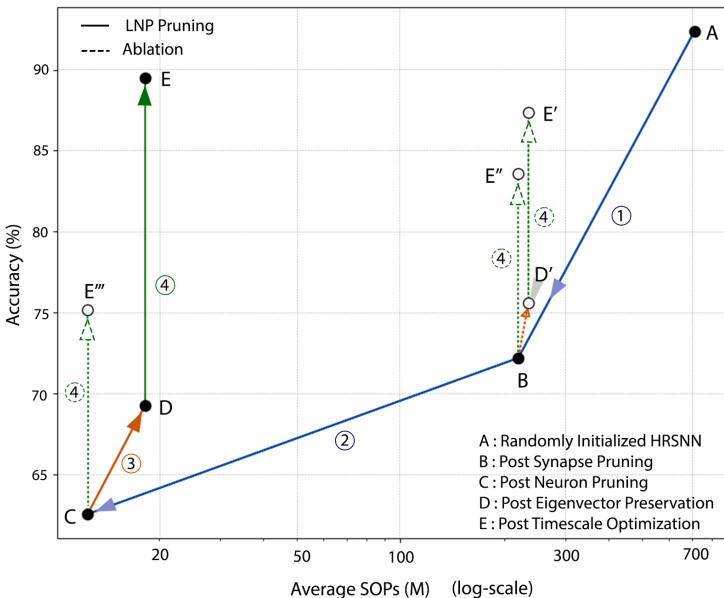

Figure 13: Plot showing Ablation studies of LNP

1. We start with Point A, which represents the randomly initialized unpruned (dense) HRSNN network

   - The blue line (A → B) indicates Step 1 of the LNP algorithm, where we prune the synapses using the noise-based pruning method.

2. Point B is the outcome of synapse pruning the randomly initialized network. There are 3 different directions we can go from here:

   - Step 2 (The standard LNP step): The blue line B → C indicates Step 2 of the LNP method, where we prune the neurons based on the betweenness centrality. This is the standard step used in the LNP algorithm.
   - Step 3: The orange line B → D' indicates Step 3 (eigenvector preservation) done directly after Step 1 (skipping Step 2 or the neuron pruning step).
     - This is followed by Step 4 (Timescale Optimization) to reach the ablation model E'. We see the performance of E' is quite high but the Avg. SOPs is much higher than E (LNP solution).
   - Step 4: We can directly optimize the neuron timescales from B (skipping steps 2,3), reaching E''. However, the performance is even worse.

3. Point C is derived by using Step 2 (neuron pruning) on Point B. This reduces the SOPs significantly, but also the performance deteriorates. We can go two ways from this point:

   - Step 3 (The standard LNP step): We can do eigenvector preservation to reach point D.
   - Step 4 (Timescale Optimization) directly (skipping Step 3)- This leads to a model with very sparse connections, but performance also takes a big hit.

4. Point D derived by using Step 3 (Eigenvector Preservation) on Point C. The model performance increases a bit, but the SOPs also increase due to the addition of new synapses in this step.

5. Point E: This is the final output of the LNP algorithm where we use Step 4 (Timescale Optimization) at point D. We see this process gives the model with the best performance and SOP.

# C SUPPLEMENTARY SECTION C

## C.1 RELATED WORKS

**Pruning Methods for DNN** Pruning methods in Deep Neural Networks (DNNs) are crucial for optimizing models, especially for deployment in resource-constrained environments. One common technique is magnitude-based pruning, where components of the network are selectively pruned based on their magnitudes, often without the need for retraining the model post-pruning Hoefler et al. (2021); Vadera & Ameen (2020). Another prevalent method is filter pruning, which focuses on pruning filters in convolutional layers to reduce the computational cost while maintaining model performance He et al. (2022); Kulkarni et al. (2022). Structured pruning is also employed, where pruning is performed in a structured manner, and the pruning rate for each layer is adaptively derived based on gradient and loss function Sakai et al. (2022). Sensitivity analysis is used to assess the effect of pruning, allowing for the identification of redundant components that can be removed with minimal impact on model accuracy Vadera & Ameen (2020). Multi-objective sensitivity pruning considers hardware-based objectives such as latency and energy consumption along with model accuracy to optimize the pruning process Sabih et al. (2022). Deep Q-learning-based methods like QLP intelligently determine favorable layer-wise sparsity ratios, implementing them via unstructured, magnitude-based, weight pruning Camci et al. (2022).

**Pruning Methods for SNN**

Pruning methods in Spiking Neural Networks (SNNs) are essential for optimizing models for deployment in resource-constrained environments. Gradient Rewiring is a method inspired by synaptogenesis and synapse elimination in the neural system, allowing for the optimization of network structure without retraining, maintaining minimal loss of SNNs' performance on various datasets. Spatio-temporal pruning is another method that utilizes principal component analysis to perform structured spatial pruning, leading to significant model compression and latency reduction Chowdhury et al. (2021a). DynSNN proposes a dynamic pruning framework that optimizes network topology on the fly, achieving almost lossless performance for SNNs on multiple datasets Liu et al. (2022a). u-Ticket addresses the workload imbalance problem in sparse SNNs by adjusting the weight connections during Lottery Ticket Hypothesis-based pruning, ensuring optimal hardware utilization Yin et al. (2023). Developmental Plasticity-inspired Adaptive Pruning (DPAP) draws inspiration from the brain's developmental plasticity mechanisms to dynamically optimize network structure during learning without pre-training and retraining, achieving efficient network architectures Han et al. (2022). Multi-strength SNNs employ an innovative deep structure that allows for aggressive pruning strategies, reducing computational operations significantly while maintaining accuracy.

**Spiking Neural Networks**

Spiking neural networks (SNNs) Ponulak & Kasinski (2011) use bio-inspired neurons and synaptic connections, trainable with either unsupervised localized learning rules such as spike-timing-dependent plasticity (STDP) Gerstner & Kistler (2002); Chakraborty et al. (2023) or supervised backpropagation-based learning algorithms such as surrogate gradient Neftci et al. (2019). SNNs are gaining popularity as a powerful low-power alternative to deep neural networks for various machine learning tasks. SNNs process information using binary spike representation, eliminating the need for multiplication operations during inference. Recent advances in neuromorphic hardware Akopyan et al. (2015), Davies et al. (2018), Kim et al. (2022a) have shown that SNNs can save energy by orders of magnitude compared to artificial neural networks (ANNs), maintaining similar performance levels. Since these networks are increasingly crucial as efficient learning methods, understanding and comparing the representations learned by different supervised and unsupervised learning models become essential. Empirical results on standard SNNs also show good performance for various tasks, including spatiotemporal data classification, Lee et al. (2017); Khoei et al. (2020), sequence-to-sequence mapping Chakraborty & Mukhopadhyay (2023a),Zhang & Li (2020), object detection Chakraborty et al. (2021); Kim et al. (2020), and universal function approximation Gelenbe et al. (1999); Iannella & Back (2001). Furthermore, recent research has demonstrated that introducing heterogeneity in the neuronal dynamics Perez-Nieves et al. (2021); Chakraborty & Mukhopadhyay (2023b; 2022); She et al. (2021) can enhance the model's performance to levels akin to supervised learning models.

**STDP-based learning in Recurrent Spiking Neural Network** Spike-Timing-Dependent Plasticity (STDP) Gerstner et al. (1993),Chakraborty & Mukhopadhyay (2021) based learning in recurrent Spiking Neural Networks (SNNs) is a biologically inspired learning mechanism that relies on the precise timing of spikes for synaptic weight adjustment. STDP enables the network to learn and generate sequences and abstract hidden states from sensory inputs, making it crucial for tasks like pattern recognition and sequence generation in recurrent SNNs. For instance, a study by Guo et al. Guo et al. (2021) proposed a supervised learning algorithm for recurrent SNNs based on BP-STDP, focusing on optimizing learning in a structured manner. Another research by van der Veen van der Veen (2022) explored the incorporation of STDP-like behavior in eligibility propagation within multi-layer recurrent SNNs, demonstrating improved classification performance in certain neuron models. Chakraborty et al. Chakraborty & Mukhopadhyay (2022) presented a heterogeneous recurrent SNN for spatio-temporal classification, utilizing heterogeneous STDP with varying learning dynamics for each synapse, showing comparable performance to backpropagation-trained supervised SNNs with less computation and training data. Panda et al. Panda & Roy (2017) combined Hebbian plasticity with a non-Hebbian synaptic decay mechanism in a recurrent spiking model to learn stable contextual dependencies between temporal sequences, suppressing chaotic activity and enhancing the model's ability to generate sequences consistently.

**Spectral Graph Sparsification Algorithm**

Spectral graph sparsification algorithms aim to reduce the complexity of graphs while preserving their spectral properties, particularly the eigenvalues of the Laplacian matrix. One such algorithm is the unweighted spectral graph sparsification algorithm, which constructs a sparsifier with fewer edges but maintains comparable graph Laplacian matrices Anderson et al. (2014). Another approach, feGRASS, focuses on scalable power grid analysis, utilizing effective edge weight and spectral edge similarity to construct low-stretch spanning trees and recover spectrally critical off-tree edges, producing spectrally similar subgraphs efficiently Liu et al. (2022c). The MAC algorithm maximizes the algebraic connectivity of sparsified measurement graphs, providing high-quality sparsification results that retain the connectivity of the graph and the quality of corresponding SLAM solutions Doherty et al. (2022). LGRASS is a linear graph spectral sparsification algorithm designed to run in strictly linear time, optimizing bottleneck subroutines and leveraging spanning tree properties Chen et al. (2022b). These algorithms are crucial for various applications, including power grid analysis, autonomous navigation, and other domains where graph analysis is essential.

# D SUPPLEMENTARY SECTION D

## D.1 BAYESIAN OPTIMIZATION FOR OPTIMIZING TIMESCALES

The majority of contemporary studies involving Bayesian Optimization (BO) predominantly focus on problems of lower dimensions, given that BO tends to perform poorly when applied to high-dimensional issues (Frazier, 2018). Nevertheless, this study endeavors to leverage BO to fine-tune the neuronal and synaptic parameters within a diverse RSNN model. This application of BO implies the optimization of a substantial array of hyperparameters, making the use of conventional BO algorithms challenging. To address this, we employ an innovative BO algorithm, predicated on the notion that the hyperparameters under optimization are correlated and follow a specific probability distribution, as indicated by Perez et al.Perez-Nieves et al. (2021). Therefore, rather than targeting individual parameters, we modify BO to approximate *parameter distributions* for LIF neurons and STDP dynamics. Once optimal distributions are identified, samples are drawn to determine the model's hyperparameter distribution. To ascertain the data's probability distribution, alterations are made to BO's surrogate model and acquisition function to focus on parameter distributions over individual variables, enhancing scalability across all dimensions. The loss for updates to the surrogate model is determined using the Wasserstein distance between parameter distributions.

BO employs a Gaussian process to represent the objective function's distribution and utilizes an acquisition function to select points for evaluation. For target dataset data points $x \in X$ and corresponding labels $y \in Y$, an SNN with network structure $\mathcal{V}$ and neuron parameters $\mathcal{W}$ serves as a function $f_{\mathcal{V},\mathcal{W}}(x)$, mapping input data $x$ to predicted label $\tilde{y}$. The optimization problem in this study is articulated as

$$\min_{\mathcal{V},\mathcal{W}} \sum_{x \in X, y \in Y} \mathcal{L}\left(y, f_{\mathcal{V},\mathcal{W}}(x)\right) \tag{11}$$

where $\mathcal{V}$ represents the neuron's hyperparameter set in $\mathcal{R}$ (Hyperparameter details are provided in the Supplementary), and $\mathcal{W}$ represents the multi-variate distribution comprising the distributions of various parameters, including the membrane time constants and the scaling function constants for the STDP learning rule in $\mathcal{S_{RR}}$.

Furthermore, BO requires a prior distribution of the objective function $f(\vec{x})$ based on the provided data $\mathcal{D}_{1:k} = \{\vec{x}_{1:k}, f(\vec{x}_{1:k})\}$. In GP-based BO, it is presumed that the prior distribution of $f(\vec{x}_{1:k})$ adheres to the multivariate Gaussian distribution, following a GP with mean $\vec{\mu}_{\mathcal{D}_{1:k}}$ and covariance $\vec{\Sigma}_{\mathcal{D}_{1:k}}$. We, therefore, calculate $\vec{\Sigma}_{\mathcal{D}_{1:k}}$ using a modified Matern kernel function, with the loss function being the Wasserstein distance between the multivariate distributions of different parameters. For higher-dimensional spaces, the Sinkhorn distance is utilized as a regularized variant of the Wasserstein distance to approximate it (Feydy et al., 2019).

$\mathcal{D}_{1:k}$ represents the points evaluated by the objective function. The GP will predict the mean $\vec{\mu}_{\mathcal{D}_{k:n}}$ and variance $\vec{\sigma}_{\mathcal{D}_{k:n}}$ for the remaining unevaluated data $\mathcal{D}_{k:n}$. The acquisition function in this study is the expected improvement (EI) of the prediction fitness as:

$$EI(\vec{x}_{k:n}) = (\vec{\mu}_{\mathcal{D}_{k:n}} - f(x_{\text{best}})) \Phi(\vec{Z}) + \vec{\sigma}_{\mathcal{D}_{k:n}} \phi(\vec{Z}) \tag{12}$$

where $\Phi(\cdot)$ and $\phi(\cdot)$ represent the probability distribution function and the cumulative distribution function of the prior distributions, respectively. BO will select the data $x_j = \text{argmax}\{EI(\vec{x}_{k:n}); x_j \subseteq \vec{x}_{k:n}\}$ as the next point for evaluation using the original objective function.

# E SUPPLEMENTARY SECTION E

## E.1 ALGORITHMS

---

**Algorithm 2** Compute Lyapunov Exponents

---

1: **Input:** Batch of input sequences, number of time steps, $n$ different input sequences.
2: **Output:** Computed Lyapunov Exponents.
3: **Initialize:** Choose a set of sequences as the validation set.
4: **for** each input sequence in the batch **do**
5:     Initialize matrix $\mathbf{Q}$ as identity matrix.
6:     Initialize hidden states $h_t$ as zeros.
7: **end for**
8: **for** each time step $t$ **do**
9:     Compute Jacobian $\mathbf{J}_t$ as:

$$[\mathbf{J}_t]_{ij} = \frac{\partial \mathbf{h}_t^j}{\partial \mathbf{h}_{t-1}^i}$$

10:     Update $\mathbf{Q}$ and Compute $QR$ decomposition:

$$\mathbf{Q}_{t+1}, \mathbf{R}_{t+1} = QR(\mathbf{J}_t \mathbf{Q}_t)$$

11: **end for**
12: **for** each $i^{th}$ vector at time step $t$ **do**
13:     Compute the $i^{th}$ LE $\lambda_i$ as:

$$\lambda_k = \frac{1}{T} \sum_{t=1}^{T} \log(r_t^k)$$

14: **end for**
15: Compute the average LE for each input in the batch.
16: Normalize and interpolate the LE spectrum to retain the shape of the largest network size.
17: **return** Lyapunov Exponents. =0

---

---

**Algorithm 3** Pruning Nodes with Lowest Betweenness Centrality

---

1: **Initialize:** $BC(v), T$

2: **Compute:** $BC(v) \forall v \in V$ using $BC(v) = \sum_{s \neq v \neq t} \frac{\sigma_{st}(v)}{\sigma_{st}}$

3: **Identify:** $P = \{v \in V : BC(v) < T\}$
4: **Prune:** $G'(V', E')$ by removing $v \in P$ and associated edges
5: **Return:** $G'(V', E') = 0$

---

**Algorithm 4** Iterative Activity Pruning

---

1: **Input:** Recurrent Spiking Neural Network (RSNN) model $M$
2: **Parameters:** Pruning rate $r$, maximum number of iterations $T$
3: **Output:** Pruned and retrained RSNN model $M'$
4: Initialize iteration counter: $t = 0$
5: Evaluate the initial performance of the unpruned model $P_{\text{initial}}$
6: **while** $t < T$ and $P_{\text{current}} \geq 0.1 \cdot P_{\text{initial}}$ **do**
7:     Evaluate the activity of each neuron in model $M$ to obtain activity list $A$
8:     Sort the neurons in $A$ in ascending order based on activity
9:     Calculate the number of neurons to prune: $n_{\text{prune}} = \text{ceil}(r \cdot \text{number of neurons in } M)$
10:     Prune the $n_{\text{prune}}$ least active neurons from model $M$
11:     Retrain the pruned model $M$ to obtain the new model with updated parameters $M'$
12:     Evaluate the performance $P_{\text{current}}$ of the model $M'$
13:     **if** $P_{\text{current}} < 0.1 \cdot P_{\text{initial}}$ **then**
14:       Break the loop to prevent further degradation in performance
15:     **end if**
16:     Update model: $M = M'$
17:     Increment iteration counter: $t = t + 1$
18: **end while**
19: **return** Pruned and retrained model $M' = 0$

---

