# OpenReview forum: "Sparse Spiking Neural Network: Exploiting Heterogeneity in Timescales for Pruning Recurrent SNN"
_ICLR.cc/2024/Conference — ICLR 2024 poster_

### Official Review · Reviewer_mdv5 · 2023-10-31

**Soundness:** 3 good
**Presentation:** 2 fair
**Contribution:** 2 fair
**Rating:** 5
**Confidence:** 4

**Summary:**

The authors significantly improved the quality of the manuscript during the revision. However, the value of the pruning strategy compared to the SOTA SNN training and quantization approach is not sufficiently significant. Based on such effort and shortcomings, I would love to increase my rating to 5.
--------------------------------------------------------
This paper proposes an innovative method called LNP, which prunes an untrained recurrent spike network via the utilization of graph theory and Lyapunov spectra. The approach proposed in this paper is better suited for discovering a remarkable sub-network in liquid-state machines or similar situations. In the SNN, energy consumption is the factor that measures the network efficiency, and not only network sparsity but also spike frequency will also affect it. Due to the variation in spike frequency before and after training, it is almost impossible to predict whether the pruned substructure remains efficient after the training process.

**Strengths:**

The LNP enables pruning the network prior to training, and effectively enhancing the sparsity while maintaining commendable performance.

**Weaknesses:**

1. The authors should elucidate the motivation behind their proposed method. What significance does their method hold for the field of spiking neural networks?
2. The network model and task employed in the experiments are too simple and do not need pruning. I am more intrigued by the adaptability of the LNP method under more intricate situations.
3. The authors provided a detailed description of the latest pruning works in SNN in the second paragraph of the Introduction. However, their results are not referenced in Table 1. Meanwhile, the compared works in Table 1 lack reference links.
4. The equation 3 appears abrupt. I don’t understand why the spike frequency of untrained SNN can be used to construct this linearized model. And how to calculate the matrix D and L?
5. Activity Pruning needs to provide citation or explanation.
6. Lorenz and Rossler are unfamiliar datasets in the field of SNN. Could you please provide the citations?

**Questions:**

See the weakness please.

---

> ### Author Response · Authors · 2023-11-20
> **Response to Reviewer mdv5 (1)**
>
> ___
> ___
> We thank the reviewer for their insightful comments and constructive feedback on our manuscript. We appreciate the opportunity to clarify and expand upon key aspects of our work. We hope we have addressed their concerns and questions regarding the paper and hope they will reconsider their rating.
> ___
> ___
>
>
> >  The authors should elucidate the motivation behind their proposed method. What significance does their method hold for the field of spiking neural networks?
>
> **>>**  We have updated the introduction to highlight the motivation for the proposed method.
>
> Spiking neural networks (SNNs), employ binary spikes for neuron communication, offering a significant energy-saving advantage in neuromorphic hardware deployment due to their event-driven operations. Recently, there has been a growing interest in developing recurrent SNN architectures with heterogeneous timescales for solving several spatio-temporal classification and temporal prediction tasks. Though introducing such heterogeneities helps improve the model performance, it exponentially increases the computational complexity of the model and makes it extremely hard to train and stabilize.
>
> To address this issue, researchers have been working on developing pruning methods to  significantly reduce the number of neurons and synaptic connections with minimal impact on performance. However, most existing pruning methods, which were initially developed for ANNs, do not fully exploit the unique characteristics of SNNs, particularly the potential for fine-grained neuron pruning. This oversight limits the full realization of the sparse computing capabilities of neuromorphic hardware. Existing pruning works have primarily been adapted from pruning methods designed for ANNs and focused more on enabling SNNs to run on hardware with limited computational resources. Consequently, these methods do not consider the stability of the model for their pruning process, which is essential for modeling a generalizable yet robust model. Moreover, all the current methods aim to optimize performance on a given task during pruning, leading to overfitting to that task. These oversights hinder the full utilization of the sparse computing capabilities and fail to leverage the heterogeneity in dynamics of the RSNN models. These problems motivate the development of new pruning methods for SNNs, particularly for RSNNs.
>
> In this paper, we propose a novel, comprehensive \textbf{task-agnostic method} pruning framework for designing sparse heterogeneous SNNs. This approach capitalizes on the overparameterization in timescales that arises from the heterogeneity in the parameters, and the sparse, event-driven nature of spiking networks. Instead of minimizing (or constraining) performance loss for a given task, our proposed Lyapunov Noise Pruning (LNP) algorithm optimizes the model structure and parameters while pruning to preserve the stability of the sparse HRSNN. This results in sparse models that are stable and flexible, and maintain performance across multiple tasks.
>
> > The network model and task employed in the experiments are too simple and do not need pruning. I am more intrigued by the adaptability of the LNP method under more intricate situations.
>
> **>>**  We thank the reviewer for pointing this out. We have extended the LNP algorithm for spiking feedforward networks and shown the performance of the pruning method when used on Spiking ResNet architectures for CIFAR10 and CIFAR100 datasets. The detailed description of the modified algorithm is described in Supple Section A.2.
> The results of the network are given as follows:
> | | CIFAR10 | | | | CIFAR100 | | | |
> |---|---|---|---|---|---|---|---|---|
> | **Model** | **Baseline Accuracy** | **Accuracy** | **Neuron Sparsity** | **Synapse Sparsity** | **Baseline Accuracy** | **Accuracy** | **Neuron Sparsity** | **Synapse Sparsity** |
> | **Spiking ResNet19** *(Random Initialization)* | 92.11±0.9 | -2.15±0.19 | 90.48 | 94.32 | 73.32±0.81 | -3.56±0.39 | 90.48 | 94.32 |
> | **Spiking ResNet19** *(Converted)* | 93.29±0.74 | -0.04±0.01 | 93.67 | 98.19 | 74.66±0.65 | -0.11±0.02 | 94.44 | 98.07 |

---

> ### Author Response · Authors · 2023-11-20
> **Response to Reviewer mdv5 (2)**
>
> > The authors provided a detailed description of the latest pruning works in SNN in the second paragraph of the Introduction. However, their results are not referenced in Table 1. Meanwhile, the compared works in Table 1 lack reference links
>
> **>>**  We thank the reviewer for pointing this out. We have updated Table 1 to include all the papers referenced in the introduction and also add the relevant citations.
>
> > The equation 3 appears abrupt. I don’t understand why the spike frequency of untrained SNN can be used to construct this linearized model. And how to calculate the matrix D and L?
>
> **>>**   We are sorry for the confusion. We have updated the paper with a more complete description of how Eq 3 is derived. We have given a more complete description of the derivation and motivation for linearizing the model around the critical points in Supplementary Section A.2.
> The equation 3 in the described method linearizes the dynamics of a spiking neural network (SNN). We break it down to understand how the spike frequency of randomly initialized SNNs is used to construct this linearized model and how to calculate the matrices $\( D \)$ and $\( L \)$. The equation 3 is a linear approximation of the dynamics of an SNN. It's presented as:
>
> $$
> \frac{d \boldsymbol{x}}{d t} = -D \boldsymbol{x} + L \boldsymbol{x} + \boldsymbol{b}(t) = A \boldsymbol{x} + \boldsymbol{b}(t)
> $$
>
> In this equation:
>
> - $\(\boldsymbol{x}\)$ represents the vector of firing rates of neurons in the network.
> - $\(D\)$ is a diagonal matrix indicating neurons' intrinsic leak or excitability.
> - $\(L\)$ is the Lyapunov matrix.
> - $\(\boldsymbol{b}(t)\)$ denotes external inputs, including biases.
> - $\(A\)$ is a matrix defined as $\(A = -D + L\)$.
>
>  The spike frequency of randomly initialized SNNs is used to approximate the firing rates (\(\boldsymbol{x}\)) in the network.  In a randomly initialized SNN, the firing rates can be considered as raw or initial responses to inputs before any learning or adaptation has occurred. This approximation is useful for constructing a linearized model as it provides a baseline from which the effects of learning, pruning, and other dynamics can be analyzed.  Each diagonal element \( D_{ii} \) of the matrix \( D \) quantifies how the firing rate of neuron \( i \) changes over time without external input or interaction. These elements can be determined based on the inherent properties of the neurons in the network, such as their leakiness or excitability. The Lyapunov matrix \( L \) encapsulates the impact of neighboring nodes on each edge regarding their Lyapunov exponents. The elements of \( L \) are calculated using the harmonic mean of the Lyapunov exponents of the neighbors of nodes \( i \) and \( j \) as detailed in your method.  This matrix represents how the dynamics of one neuron affect its neighbors, influenced by the network's overall stability and dynamical behavior.
> In summary, the linearized model provided by equation 3 is a simplification that helps to understand the fundamental dynamics of the SNN. It uses the initial spike frequencies to establish a baseline for the network's behavior, and the matrices \( D \) and \( L \) are calculated based on the intrinsic properties of the neurons and their interactions, respectively.
>
>
> > Activity Pruning needs to provide citation or explanation.
>
> **>>**  We have added some references and a section in Supplementary Section A.3 detailing the entire process and algorithm for the activity pruning method.

---

> ### Author Response · Authors · 2023-11-20
> **Response to Reviewer mdv5 (3)**
>
> > Lorenz and Rossler are unfamiliar datasets in the field of SNN. Could you please provide the citations?
>
> **>>**  We have added citations to the Lorenz and Rossler datasets and also added a complete discussion of these dynamical systems in Supplementary Section A.4. In addition to this. we have also evaluated the model performance on more common datasets like CIFAR10 and CIFAR100 datasets. The result is shown in Table 1
>
> | **Method**                       | **Model**            | **Baseline Accuracy** | **Accuracy Loss** | **Neuron Sparsity** | **Synapse Sparsity** | **SOP Ratio** | **Baseline Accuracy** | **Accuracy Loss** | **Neuron Sparsity** | **Synapse Sparsity** | **SOP Ratio** |
> |----------------------------------|----------------------|-----------------------|-------------------|---------------------|----------------------|---------------|-----------------------|-------------------|---------------------|----------------------|---------------|
> |                                  |                      | *CIFAR10*             |                   |                     |                      |               | *CIFAR100*            |                   |                     |                      |               |
> | *ADMM [[deng2021comprehensive]]* | 7Conv, 2FC           | 89.53                 | -3.85             | -                   | 90                   | 2.91          | -                     | -                 | -                   | -                    | -             |
> | *Bellec et al [[bellec2018long]]* | 6Conv, 2FC           | 92.84                 | -3.53             | -                   | 97.96                | 16.59         | -                     | -                 | -                   | -                    | -             |
> | *Grad R [[chen2021pruning]]*      | 6Conv, 2FC           | 92.54                 | -0.30             | -                   | 71.59                | 2.09          | 71.34                 | -4.03             | -                   | 97.65                | 19.45         |
> | *IMP [[kim2022exploring]]*        | Spiking ResNet19             | 93.22                 | -0.04             | -                   | 97.54                | 13.29         | 71.34*                | -2.39*            | -                   | 97.54                | 18.69         |
> | *STDS [[chen2022state]]*          | 6Conv, 2FC           | 92.49                 | -0.35             | -                   | 88.67                | 5.27          | -                     | -                 | -                   | -                    | -             |
> | *ESL-SNN [[shen2023esl]]*         | Spiking ResNet19             | 91.09                 | -1.70             | -                   | 95                   | 2.11          | 73.48                 | -0.99             | -                   | 95                    | 14.22         |
> | **LNP (ours)**                    | Spiking ResNet19 (Randomly Initialized) | 93.29 ± 0.74          | -2.15 ± 0.19      | 90.48               | 94.32                | 9.36 ± 1.28   | 73.32 ± 0.81          | -3.96 ± 0.39      | 90.48               | 94.32                | 12.04 ± 0.41  |
> |                                  | Spiking ResNet19 (converted) |                       | -0.04 ± 0.01      | 93.67               | 98.19                | 22.18 ± 0.20  |                       | -0.11 ± 0.02      | 94.44               | 98.07                | 31.15 ± 0.28  |
> |                                  | HRSNN-STDP           | 90.26 ± 0.88          | -0.76 ± 0.07      | 94.03               | 95.06                | 39.01 ± 0.47  | 68.94 ± 0.73          | -2.16 ± 0.28      | 92.02               | 94.21                | 48.21 ± 0.59  |
> |                                  | HRSNN-BP             | 92.37 ± 0.91          | -0.81 ± 0.08      | 94.03               | 95.06                | 35.67 ± 0.41  | 70.12 ± 0.71          | -2.37 ± 0.32      | 92.02               | 94.21                | 42.57 ± 0.63  |
> |                                  | CHRSNN-STDP          | 91.58 ± 0.83          | -0.62 ± 0.07      | 95.04               | 96.68                | 50.37 ± 0.61  | 69.96 ± 0.68          | -1.65 ± 0.21      | 93.47               | 97.04                | 57.44 ± 0.68  |
> |                                  | CHRSNN-BP            | 93.45 ± 0.87          | -0.74 ± 0.07      | 95.04               | 96.68                | 45.32 ± 0.58  | 73.45 ± 0.66          | -1.11 ± 0.23      | 93.47               | 97.04                | 50.35 ± 0.64  |

---

### Official Review · Reviewer_YTsp · 2023-10-31

**Soundness:** 3 good
**Presentation:** 3 good
**Contribution:** 3 good
**Rating:** 8
**Confidence:** 4

**Summary:**

In this study, the task-agnostic Lyapunov Noise pruning (LNP) on Heterogeneous RSNN (HRSNN) is first proposed, and designed in a graph sparsification manner. HRSNN has a similar hierarchical structure as the Liquid State Machine (LSM), which includes a subsequent readout layer. LNP includes four steps 1) random dropout in HRSNN synaptic weights, where dropout rate is determined by the Lyapunov exponent, 2) mask out neurons with the lowest betweenness centrality from HRSNN, 3) add edges to maximize degree heterogeneity within local neighborhoods, 4) a Bayesian optimization for fine-tuning hyperparameter set of neurons. Results show that LNP maintains performance on time-series prediction on two chaotic systems, and two real-world datasets when FLOPS/#parameters are reduced.

**Strengths:**

LNP consider the equivalence between dense and sparse network, rather than simply optimizing the performance, which makes it suitable for random graph model like HRSNN. The design of LNP makes it not only task-agnostic but also learning-rule-agnostic, endowing SNNs from distinct research backgrounds with abundant flexibility in choice. Stability in pruning is also an interesting topic considering that SNN is a dynamic system.

**Weaknesses:**

As an SNN pruning algorithm, there are still some vague parts influencing my rate of score:

1. The calculation of sparsity is unclear to me. In most previous SNN pruning studies, the parameter/neuron sparsity includes all learnable parameters and should also include readout layers here. This vagueness could lead to bias in evaluating the power of the pruning algorithm.
2. The results on CIFAR-10 display no error bar. Considering that the spectral graph pruning includes a random dropout, the accuracy variance should also be reported.
3. LNP is essentially composed of four individual and loosely related parts with different goals. None of those are indispensable for one another. The authors should exhibit an ablation study on the contribution of each component.

**Questions:**

Considering that non-recurrent SNNs are still popular among SNN communities, is it possible to generalize and experiment on purely feed-forward SNNs like previous works do?

---

> ### Author Response · Authors · 2023-11-20
> **Response to Reviewer YTsp (1)**
>
> ___
> ___
> We thank the reviewer for their insightful comments and constructive feedback on our manuscript. We appreciate the opportunity to clarify and expand upon key aspects of our work. We hope we have addressed their concerns and questions regarding the paper and hope they will reconsider their rating.
> ___
> ___
>
> >  The calculation of sparsity is unclear to me. In most previous SNN pruning studies, the parameter/neuron sparsity includes all learnable parameters and should also include readout layers here. This vagueness could lead to bias in evaluating the power of the pruning algorithm.
>
> **>>**  We thank the reviewer for pointing this out. We apologize for the confusion. We have updated the values of sparsity in Table 1 to include the readout layer.
>
> As mentioned in the paper, the read-out layer is task-dependent and uses supervised training. In this paper, we do not explicitly prune the read-out network, but the read-out network is implicitly pruned. The readout layer is a multi-layer (two or three layers) fully connected network. The first layer size is equal to the number of neurons sampled from the recurrent layer. We sample the top 10% of neurons with the greatest betweenness centrality. Thus, as the number of neurons in the recurrent layer decreases, the size of the first layer of the read-out network also decreases. The second layer consists of fixed 20 neurons. The third layer differs between the classification and the prediction tasks such that for classification, the number of neurons in the third layer is equal to the number of classes. On the other hand, the third layer for the prediction task is a single neuron which gives the prediction output. We have updated the values of sparsity in Table 1 to include the readout layer.
>
> |        | Neuron  Sparsity Change | Synapse Sparsity Change | Avg SOP change CIFAR10 | Avg SOP change CIFAR100 |
> |:------:|:-----------------------:|:-----------------------:|:----------------------:|:-----------------------:|
> |  HRSNN |          -0.13          |          -0.18          |          -0.09         |          -0.14          |
> | CHRSNN |          -0.08          |          -0.13          |          -0.1          |          -0.16          |
>
>
> > The results on CIFAR-10 display no error bar. Considering that the spectral graph pruning includes a random dropout, the accuracy variance should also be reported.
>
> **>>**  We have added a variance term in the results shown in Table 1 for CIFAR 10 and CIFAR 100 by repeating each of the experiments 10 times. We see that the CHRSNN still shows better performance with slightly higher SOPs than HRSNN models.
>
> | **Method** | **Model** | **Baseline Accuracy (CIFAR10)** | **Accuracy Loss (CIFAR10)** | **Neuron Sparsity (CIFAR10)** | **Synapse Sparsity (CIFAR10)** | **SOP Ratio (CIFAR10)** | **Baseline Accuracy (CIFAR100)** | **Accuracy Loss (CIFAR100)** | **Neuron Sparsity (CIFAR100)** | **Synapse Sparsity (CIFAR100)** | **SOP Ratio (CIFAR100)** |
> |------------|-----------|---------------------------------|-----------------------------|-------------------------------|--------------------------------|-------------------------|----------------------------------|------------------------------|--------------------------------|---------------------------------|--------------------------|
> | **LNP (ours)** | Spiking ResNet19 (Random Initialization) | 93.29 ± 0.74 | -2.15 ± 0.19 | 90.48 | 94.32 | 9.36 ± 1.28 | 73.32 ± 0.81 | -3.96 ± 0.39 | 90.48 | 94.32 | 12.04 ± 0.41 |
> |  | Spiking ResNet19 (converted) | 93.29 ± 0.74 | -0.04 ± 0.01 | 93.67 | 98.19 | 22.18 ± 0.2 | - | -0.11 ± 0.02 | 94.44 | 98.07 | 31.15 ± 0.28 |
> |  | HRSNN-STDP | 90.26 ± 0.88 | -0.76 ± 0.07 | 94.03 | 95.06 | 39.01 ± 0.47 | 68.94 ± 0.73 | -2.16 ± 0.28 | 92.02 | 94.21 | 48.21 ± 0.59 |
> |  | HRSNN-BP | 92.37 ± 0.91 | -0.81 ± 0.08 |  94.03 | 95.06  | 35.67 ± 0.41 | 70.12 ± 0.71 | -2.37 ± 0.32 |  92.02 | 94.21  | 42.57 ± 0.63 |
> |  | CHRSNN-STDP | 91.58 ± 0.83 | -0.62 ± 0.07 | 95.04 | 96.68 | 50.37 ± 0.61 | 69.96 ± 0.68 | -1.65 ± 0.21 | 93.47 | 97.04 | 57.44 ± 0.68 |
> |  | CHRSNN-BP | 93.45 ± 0.87 | -0.74 ± 0.07 |  95.04 | 96.68  | 45.32 ± 0.58 | 73.45 ± 0.66 | -1.11 ± 0.23 | 93.47 | 97.04 | 50.35 ± 0.64 |

---

> ### Author Response · Authors · 2023-11-20
> **Response to Reviewer YTsp (2)**
>
> > LNP is essentially composed of four individual and loosely related parts with different goals. None of those are indispensable for one another. The authors should exhibit an ablation study on the contribution of each component.
>
> **>>**   We conducted an ablation study, where we systematically examined various combinations of the four sequential steps involved in the LNP method. This study's findings are presented graphically, as illustrated in Fig. 5. At each point (A-E), we train the model and obtain the model's accuracy and average. Synaptic Operations (SOPs) to support ablation studies. The ablation study is done for the HRSNN model, which is trained using STDP and evaluated on the CIFAR10 dataset. In the figure, different line styles and colors represent distinct aspects of the procedure: the blue line corresponds to Steps 1 and 2 of the LNP process, the orange line to Step 3, and the green line to Step 4. Solid lines depict the progression of the original LNP process $(A \rightarrow B \rightarrow C \rightarrow D \rightarrow E)$, while dotted lines represent the ablation studies conducted at stages B and C. This visual representation enables a clear understanding of the individual impact each step exerts on the model's performance and efficiency. Additionally, it provides insights into potential alternative outcomes that might have arisen from employing different permutations of these steps or omitting certain steps altogether.
>
>  For this figure, we consider the HRSNN network, which is trained using STDP and evaluated on the CIFAR10 dataset. Below is a detailed discussion of the figure:
>
> For this figure, we consider the HRSNN network, which is trained using STDP and evaluated on the CIFAR10 dataset. Below is a detailed discussion of the figure:
>
> 1. **Point A - Initial State:**
>    - The randomly initialized unpruned (dense) HRSNN network is represented by Point A.
>    - The blue line ($A \rightarrow B$) indicates Step 1 of the LNP algorithm, where we prune the synapses using the noise-based pruning method.
>
> 2. **Point B - Outcome of Synapse Pruning:**
>    - Point B is the outcome of synapse pruning the randomly initialized network. From here, there are 3 different directions:
>      - **Step 2 (Standard LNP Step):**  *[The originalLNP step]*
>        - The blue line ($B \rightarrow C$)  represents Step 2 of the LNP method, where we prune neurons based on betweenness centrality. This is the standard step used in the LNP algorithm.
>      - **Step 3 (Eigenvector Preservation):**
>        - The orange line ($B \rightarrow D'$)  indicates Step 3, done directly after Step 1 (skipping Step 2). This is followed by Step 4 (Timescale Optimization) to reach the ablation model E’. The performance of E’ is high, but the Avg. SOPs are much higher than E (LNP solution).
>      - **Step 4 (Timescale Optimization):**
>        - Optimizing neuron timescales directly from B (skipping steps 2 and 3) leads to E”. The performance in this scenario is worse.
>
> 3. **Point C - Neuron Pruning:**
>    - Derived by using Step 2 (neuron pruning) on Point B. This significantly reduces SOPs but also deteriorates performance. Two pathways are possible from this point:
>      - **Step 3 (Eigenvector Preservation):** *[The originalLNP step]*
>        - Performing eigenvector preservation reaches point D.
>      - **Step 4 (Timescale Optimization) Directly:**
>        - This leads to a model with very sparse connections, but performance also suffers greatly.
>
> 4. **Point D - Post-Eigenvector Preservation:**
>    - Derived by using Step 3 (Eigenvector Preservation) on Point C. The model performance increases slightly, but SOPs also increase due to the addition of new synapses in this step.
>
> 5. **Point E - Final Output of LNP Algorithm:**
> - This is the final output of the original LNP algorithm
>    - Achieved by using Step 4 (Timescale Optimization) at point D. This process yields the model with the best performance and SOP balance.
>
>
>
> [Plot showing Ablation studies of LNP on HRSNN trained with STDP on CIFAR10](https://anonymous.4open.science/r/ICLR24-1FD2/Ablation_v2.png)

---

> ### Author Response · Authors · 2023-11-20
> **Response to Reviewer YTsp (3)**
>
> > Considering that non-recurrent SNNs are still popular among SNN communities, is it possible to generalize and experiment on purely feed-forward SNNs like previous works do?
>
> **>>**   We would like to thank the reviewer for pointing this out. We have extended the algorithm for use on Feedforward Spiking Neural Networks like Spiking ResNets for CIFAR10 and CIFAR100 datasets. The results are added in Table 1 and Fig 3. We have also added the description of the algorithm for the Feedforward Spiking networks in Supplementary Section A.2. A short description of the changes for the feedforward Spiking networks is given below
>
> 1.	For the structured pruning in Step 2, instead of pruning neurons, we prune channels
> 2.	For Step 3, we use norm preservation using skip connections
>
> ### Table showing the performance of LNP on Feedforward Spiking Neural Networks
>
> | | CIFAR10 | | | | CIFAR100 | | | |
> |---|---|---|---|---|---|---|---|---|
> | **Model** | **Baseline Accuracy** | **Accuracy** | **Neuron Sparsity** | **Synapse Sparsity** | **Baseline Accuracy** | **Accuracy** | **Neuron Sparsity** | **Synapse Sparsity** |
> | **Spiking ResNet19** *(Random Initialization)* | 92.11±0.9 | -2.15±0.19 | 90.48 | 94.32 | 73.32±0.81 | -3.56±0.39 | 90.48 | 94.32 |
> | **Spiking ResNet19** *(Converted)* | 93.29±0.74 | -0.04±0.01 | 93.67 | 98.19 | 74.66±0.65 | -0.11±0.02 | 94.44 | 98.07 |
>
>
> The detailed description of the algorithm for the Feedforward Spiking networks and the results for the Spiking ResNet models on CIFAR10 and CIFAR100 are given in Table 7 in Supplementary Section B.2

---

> ### Comment · Reviewer_YTsp · 2023-12-05
>
> I confirm that the newly conducted trials by the authors have resolved my concerns about the ablation study and randomness. In my opinion, the ablation study highlights the need for further discussion on post eigenvector preservation, while the other three strategies are necessary for achieving a better balance between SOPs and accuracy. I hope that the authors will have enough space to refine the empirical studies in the camera-ready version. Therefore, I have decided to raise my score to 8.

---

### Official Review · Reviewer_udA5 · 2023-11-03

**Soundness:** 3 good
**Presentation:** 3 good
**Contribution:** 3 good
**Rating:** 8
**Confidence:** 4

**Summary:**

This paper presents a novel task-agnostic method called Lyapunov Noise Pruning (LNP) for designing sparse heterogeneous Recurrent Spiking Neural Networks (HRSNNs). Unlike conventional approaches that prune dense networks after training them for a specific task, LNP starts with an untrained and arbitrarily initialized dense HRSNN model. It uses the Lyapunov spectrum and spectral graph sparsification algorithms to prune synapses and neurons while maintaining network stability. The resulting untrained sparse HRSNN can then be trained for various target tasks using supervised or unsupervised methods.

**Strengths:**

1. In this article, authors introduced a novel method for task-agnostic pruning, achieved by pruning a Recurrent Spiking Neural Network (RSNN) prior to the training process.

2. The study demonstrates that Lyapunov Noise Pruning (LNP) outperforms AP in terms of network efficiency enhancement.

3. This research proposes a technique that capitalizes on the heterogeneous timescales of RSNN, offering both biological plausibility and a supportive role in the pruning process.

**Weaknesses:**

1. The author did not show the difference between random initialization and LNP, and didnot analyze the dynamic property of the hetergeneous timescale of LNP and how it influence the training process of RSNN.

2. The author didnot carefully organize the citation and references. For example,  in Reference section, 'Yanqi Chen, Zhaofei Yu, Wei Fang, Tiejun Huang, and Yonghong Tian. Pruning of deep spiking neural networks through gradient rewiring. arXiv preprint arXiv:2105.04916, 2021a.', and 'Yanqing Chen, Zhaofei Yu, Wei Fang, Tiejun Huang, and Yonghong Tian. Pruning of deep spiking neural networks through gradient rewiring. 2021b. doi: 10.24963/ijcai.2021/236. URL https://doi.org/10.24963/ijcai.2021/236.' are same paper, but sited twice.

**Questions:**

1. In the study conducted by Panda et al., it was demonstrated that their approach resulted in a higher degree of sparsity and a lower impact on accuracy when compared to the LNP method. Could you please elucidate the underlying reasons for this observed outcome?

2. Could you provide a comprehensive description of the HRSNN architecture? It appears that there is a readout layer characterized by dense connections that are not subjected to pruning via the LNP technique.

3. The Lyapunov spectrum is customarily analyzed in continuous dynamical systems, such as RNNs. How did the researchers address the discontinuity of spikes in RSNNs when determining the Lyapunov spectrum?

4. Kindly elucidate the principal dynamic characteristic of LNP RSNNs prior to the completion of the final training process and provide an explanation as to why randomly sparse Initialized RSNNs lack this particular dynamical property. Could you also discuss dynamic property of LNP-RSNN, for example, using temporal correlation function and the Lyapunov spectrum in this context?

---

> ### Author Response · Authors · 2023-11-20
> **Response to Reviewer udA5 (1)**
>
> ___
> ___
> We thank the reviewer for their insightful comments and constructive feedback on our manuscript. We appreciate the opportunity to clarify and expand upon key aspects of our work. We hope we have addressed their concerns and questions regarding the paper and hope they will reconsider their rating.
> ___
> ___
>
> > The author did not show the difference between random initialization
>
> **>>**  We added the results for random initialization for each step of the iteration of the LNP - initialized 10 different networks, trained them, and showed their performance in Fig 3(c). For the plot, after each iteration, we have randomly created a network with an equal number of synapses and edges as found by the LNP method and trained and tested it on the Lorenz63 dataset to get the RMSE vs Iteration graph shown in Fig 3c (below). We see that the Random initialized network shows higher variance and shows more jumps, signifying the randomly initialized model is unstable without proper finetuning. In addition, we see the performance consistently getting worse as the model size keeps decreasing, signifying an optimal network architecture is more crucial for smaller networks than for larger networks.
>
> [Comparative Evaluation of Pruning Methods Across Iterations. ](https://anonymous.4open.science/r/ICLR24-1FD2/Iterations.pdf)
>
>
> > The author didnot carefully organize the citation and references. For example, in Reference section, 'Yanqi Chen, Zhaofei Yu, Wei Fang, Tiejun Huang, and Yonghong Tian. Pruning of deep spiking neural networks through gradient rewiring. arXiv preprint arXiv:2105.04916, 2021a.', and 'Yanqing Chen, Zhaofei Yu, Wei Fang, Tiejun Huang, and Yonghong Tian. Pruning of deep spiking neural networks through gradient rewiring. 2021b. doi: 10.24963/ijcai.2021/236. URL https://doi.org/10.24963/ijcai.2021/236.' are same paper, but sited twice.
>
> **>>**  We thank the reviewer for pointing this out. We have cleaned the bibtex to avoid such redundancies to the best of our abilities.

---

> ### Author Response · Authors · 2023-11-20
> **Response to Reviewer udA5 (2)**
>
> > In the study conducted by Panda et al., it was demonstrated that their approach resulted in a higher degree of sparsity and a lower impact on accuracy when compared to the LNP method. Could you please elucidate the underlying reasons for this observed outcome?
>
> **>>**  This is an interesting observation. The primary reason is that we have presented a task-agnostic pruning method. Our approach starts with an un-trained dense network, and the pruning process does not consider task performance while removing network connections (and neurons) and neuronal time scales. Consequently, the end-pruned network does not overfit to any particular dataset; instead, it is generalizable to many different datasets and even different tasks.
>
> On the other hand, prior SNN pruning papers (such as the one referred by the reviewer from Panda et al.) start with a pre-trained DNN model on a given task, which is converted to an SNN. During pruning, the network is simultaneously pruned and re-trained with the dataset (at each pruning step) to minimize performance loss on that dataset. This makes the model overfitted (hence, shows good results) to that dataset.
>
> We use the following terminology for the different pruning cases:
> a. *Randomly Initialized SNN:* The SNN model is not trained before pruning, and the parameters are randomly initialized. Only the final pruned network is then trained.
> b. *Converted SNN:*  A standard DNN was trained on a dataset and then converted to an SNN with the same architecture using DNN-to-SNN conversion methods.
>
> To verify our conjecture, we empirically study the following questions:
>
> 1. **How does our pruning method perform on a converted SNN model like prior works?**
>
> a. *Our Approach 1 (Spiking ResNet converted)*: We consider two cases. First, we use our pruning method on a dense Spiking ResNet, which is converted from a DNN ResNet (trained on the CIFAR10 and CIFAR100 datasets). We continue with our approach where the pruning iterations do not consider task performance. We observe that when starting from a pre-trained dense model like prior works, the performance and sparsity of the network pruned with LNP are better than prior pruning methods.
>
> b. *Our Approach 2 (Spiking ResNet Random Initialization)*: We start with a randomly initialized Spiking ResNet with randomly initialized weights and parameters without training the model at any iteration of the pruning process. The final pruned network with optimized hyperparameters (optimized using the Lyapunov spectrum) is then trained and tested on the CIFAR10 and CIFAR100 datasets.
>
> c. *Approach by IMP (Panda et al.)*: Iterative Magnitude Pruning (IMP) is a neural network compression technique involving alternating between pruning and retraining steps. In this process, a small percentage of the least significant weights are pruned, and the network is then fine-tuned to recover performance. This iterative cycle continues until a desired sparsity level is reached, potentially enhancing network efficiency with minimal impact on accuracy.
>
> 2. **How do other task-dependent pruning methods perform when the network pruned for one task is used for a different task?**
>
> To compare this generalizability property, we took the pruned model from Panda et al. paper, which is optimized for CIFAR10. We next train their pruned model on CIFAR100 sparsity, accuracy, and SOPs. We compare the performance with our pruned randomly initialized Spiking ResNet model (with random weights) and pruned converted Spiking ResNet (as described earlier). We see that the randomly initialized Spiking ResNet generated by our LNP-based pruning can better generalize to the CIFAR100 dataset and perform better. Moreover, using the converted Spiking ResNet, we see better performance and efficiency of the pruned Spiking ResNet (converted) than the IMP-based pruning model on the CIFAR10 and CIFAR100 datasets.
>
>
> |   Pruning  Method  |                 Architecture                 | Neuron Sparsity | Synapse  Sparsity | CIFAR10 Accuracy | CIFAR100 Accuracy |
> |:------------------:|:--------------------------------------------:|:---------------:|:-----------------:|:----------------:|:-----------------:|
> | IMP (Panda et al.) | Spiking  ResNet19 (iterative pruning  and retraining) |        -        |       97.54       |       93.18      |       68.95       |
> |     LNP (Ours)     |            Spiking  ResNet19  (Random Initialization)            |      90.48      |       94.32       | 89.96 $\pm$ 0.71 |  69.76 $\pm$ 0.42 |
> |     LNP (Ours)     |            Spiking ResNet  (converted)           |      93.67      |       98.19       | 93.25 $\pm$ 0.73 |  74.55 $\pm$ 0.63 |

---

> ### Author Response · Authors · 2023-11-20
> **Response to Reviewer udA5 (3)**
>
> > Could you provide a comprehensive description of the HRSNN architecture? It appears that there is a readout layer characterized by dense connections that are not subjected to pruning via the LNP technique.
>
> **>>**  We have updated the paper with a more detailed description of the HRSNN model and the details are added in Supplementary Section A.7.
>
> Regarding the readout layer, we are sorry for the confusion and would like to thank the reviewer for pointing that out. As mentioned in the paper, the read-out layer is task-dependent and use supervised training. In this paper, we do not explicitly prune the read-out network, but the readout network is implicitly pruned. The readout layer is a multi-layer (two or three layers) fully connected network. The first layer size is equal to the number of neurons sampled from the recurrent layer. We sample the top 10% of neurons with the greatest betweenness centrality. Thus, as the number of neurons of the recurrent layer decreases, the size of the first layer of the read-out network also decreases. The second layer consists of fixed 20 neurons. The third layer differs between the classification and the prediction tasks such that for classification, the number of neurons in the third layer is equal to the number of classes. On the other hand, the third layer for the prediction task is a single neuron which gives the prediction output. We have updated the values of sparsity in Table 1 to include the readout layer. We see a small dip in the neuron and sparsity levels and the avg. SOPs when we consider the readout layer.  The results are shown below. The details are added in Suppl Sec A.7
>
> |        | Neuron  Sparsity Change | Synapse Sparsity Change | Avg SOP change CIFAR10 | Avg SOP change CIFAR100 |
> |:------:|:-----------------------:|:-----------------------:|:----------------------:|:-----------------------:|
> |  HRSNN |          -0.13          |          -0.18          |          -0.09         |          -0.14          |
> | CHRSNN |          -0.08          |          -0.13          |          -0.11          |          -0.17          |

---

> ### Author Response · Authors · 2023-11-20
> **Response to Reviewer udA5 (4)**
>
> > The Lyapunov spectrum is customarily analyzed in continuous dynamical systems, such as RNNs. How did the researchers address the discontinuity of spikes in RSNNs when determining the Lyapunov spectrum?
>
> **>>**  We follow the principles outlined by Engelken et al. [1] for calculating the Lyapunov spectrum of the discrete-time firing-rate network.
> Conducting numerical simulations with small perturbations in Recurrent Spiking Neural Networks (RSNNs), particularly in the context of discrete spiking events, requires a specific approach to accurately capture these networks' dynamics.
> 1. **Network Initialization**: Set up an RSNN with a defined architecture, synaptic weights, and initial neuronal states.
>
> 2. **Creating Perturbations**: Generate a slightly perturbed version of the network. This could involve minor adjustments to the initial membrane potentials or other state variables of a subset of neurons.
>
> 3. **Simulating Network Dynamics**:  Run parallel simulations of the original and the perturbed RSNNs, ensuring they receive identical input stimuli.
>    - Since the networks operate on discrete spike events, the state of each neuron is updated based on the inputs it receives and its current potential
>
> 4. **Measuring Divergence**:  At each time step, measure the difference between the states of the two networks. In the context of spiking neurons, this could involve comparing the spike trains of corresponding neurons in each network. This difference could be quantified using various metrics, such as spike-timing difference, spike count difference, or membrane potential differences.
>
> 5. **Tracking the Evolution of the Perturbation**: Observe how the initial small differences evolve over time. These differences might lead to significantly divergent spiking patterns in a network exhibiting chaotic dynamics.
>
> 6. **Estimating Lyapunov Exponents**: Calculate the rate at which the trajectories of the original and perturbed networks diverge. This involves fitting an exponential curve to the divergence data over time. The slope of this curve gives an estimate of the Lyapunov exponent. A positive exponent indicates sensitivity to initial conditions and potential chaotic dynamics.
>
> The evolution of the map is given by
> $$
> h_i\left(t_s+\Delta t\right)=f_i=(1-\Delta t) h_i\left(t_s\right)+\Delta t \sum_{j=1}^N J_{i j} \phi\left(h_j\left(t_s\right)\right)
> $$
> In the limit $\Delta t \rightarrow 0$, a continuous-time dynamics \cite{} is recovered. For $\Delta t=1$, the discrete-time network \cite{} is obtained.
> The Jacobian for the discrete-time map is
>
> $$
> D_{i j}\left(t_s\right) = \left.\frac{\partial f_i}{\partial h_j}\right|_{t=t_s} = (1-\Delta t) \delta + \Delta t \cdot J \phi^{\prime}\left(h_j\left(t_s\right)\right) .
> $$
>
> The full Lyapunov spectrum is again obtained by a reorthonormalization procedure of the Jacobians along a numerical solution of the map; for details, see Supplemental Section A.1
>
>
>
> ## References:
>
> [1] Engelken, R., Wolf, F. and Abbott, L.F., 2023. Lyapunov spectra of chaotic recurrent neural networks. Physical Review Research, 5(4), p.043044.

---

> ### Author Response · Authors · 2023-11-20
> **Response to Reviewer udA5 (5)**
>
> > Kindly elucidate the principal dynamic characteristic of LNP RSNNs prior to the completion of the final training process and provide an explanation as to why randomly sparse Initialized RSNNs lack this particular dynamical property. Could you also discuss dynamic property of LNP-RSNN, for example, using temporal correlation function and the Lyapunov spectrum in this context?
>
> >>  (The authors) didnot analyze the dynamic property of the hetergeneous timescale of LNP and how it influence the training process of RSNN
>
> **>>**   We express our sincere gratitude to the reviewer for their valuable suggestion. The proposed topic, focusing on exploring the dynamical characteristics of heterogeneous RSNN models, presents an intriguing and expansive area of study. We fully acknowledge its significance and are enthusiastic about conducting a more in-depth investigation in our future works.
>
> In this paper, we have concentrated on the empirical study of the Lyapunov spectrum, a modification made in response to the reviewer's insightful recommendation. We believe that this addition not only enriches the current paper but also opens up numerous avenues for future research, enhancing the paper's overall contribution to the field.
>  To show the principal dynamic characteristic of the LNP-HRSNN model, we plot the full Lyapunov spectrum of the HRSNN model for three different cases - the unpruned network, the pruned network, and the trained pruned network. We refer to the methodologies discussed in recent works [1,2] The Lyapunov Spectrum provides valuable additional insights into the collective dynamics of firing-rate networks. We plot the evolution of Lyapunov spectra with different initialization parameters. We plotted the Lyapunov spectrum, with three different probabilities of synaptic connection for the initial network (p=0.001, p=0.01, p=0.1). We plot the Lyapunov exponents ($\lambda_i$) vs the normalized indices $i/N$ described as follows:
>
> - **$\lambda_i$**: This axis represents the Lyapunov exponents. A positive exponent indicates chaos, meaning that two nearby trajectories in the phase space will diverge exponentially. A negative exponent suggests that trajectories converge, and zero would imply neutral stability.
>
> - **i/N**: This axis is likely indexing the normalized Lyapunov exponents, with \( i \) being the index of a particular exponent and \( N \) being the total number of exponents calculated. For a system with \( N \) dimensions, there are \( N \) Lyapunov exponents.
>
> The graph shows three plots of the Lyapunov spectrum for different stages of pruning and training:
>
> 1. **Unpruned**: This plot represents the Lyapunov spectrum of the neural network before any pruning has been done. The spectrum shows a range of Lyapunov exponents from positive to negative values, indicating that the network likely has both stable and chaotic behaviors. More notably, the more sparse initialized models (p = 0.1) were more unstable as the largest Lyapunov exponent is positive
>
> 2. **Pruned Pre-Training**: This plot shows the Lyapunov spectrum after the network has been pruned but before it has been trained again. Pruning is a process in which less important connections (weights) in a neural network are removed, which can simplify the network and potentially lead to more efficient operation without significantly impacting performance.
>
> 3. **Pruned Post-Training**: This plot illustrates the Lyapunov spectrum after the network has been pruned and then trained again. Retraining the network after pruning allows it to adjust the remaining weights to compensate for the removed connections, which can restore or even improve performance.
>
> [Lyapunov Spectrum of the HRSNN model in three stages of pruning and training and different levels of sparsity for initialization: (a) the Lyapunov Spectrum of unpruned HRSNN model with a probability of connection $p=0.1, 0.01, 0.001$ (b) the Lyapunov Spectrum after LNP pruning (c) the Lyapunov Spectrum after training of the pruned model](https://anonymous.4open.science/r/ICLR24-1FD2/lyapunov_updated.png)

---

> ### Author Response · Authors · 2023-11-20
> **Response to Reviewer udA5 (5 - Contd.)**
>
> We can infer the following from the plots:
>
> - **Impact of Pruning**: Comparing the 'Unpruned' to the 'Pruned Pre-Training' plot, it seems that pruning increases the stability of the system as the Lyapunov exponents become more negative (less chaotic). This might suggest that pruning reduces the complexity of the dynamics.
>
> - **Training Effect**: The 'Pruned Post-Training' plot shows that after re-training, the exponents are less negative compared to 'Pruned Pre-Training,' which may indicate that the network is able to regain some dynamical complexity or expressive power through training.
>
> - **Initialization Synapse Density**: Different initialization synapse densities (arising from different p) have different effects on the spectrum. High \( p \)) tends to result in more negative exponents, suggesting greater stability but potentially less capacity to model complex patterns.
>
> - **Network Robustness**: The fact that the Lyapunov spectrum does not dramatically change in shape across the three stages, especially in the 'Pruned Post-Training' stage, might imply that the network is robust to pruning, retaining its general dynamical behavior while likely improving its efficiency.
>
> - **Convergence of Spectra:** The convergence of the spectra for different values of
> p after pruning, both pre-and post-training, suggests that regardless of the initial density of connections, the network may evolve towards a structure that is similarly efficient for the task it is being trained on. This indicates that the LNP algorithm can efficiently prune a network irrespective of the initial density and also make the final model more stable.
>
> Overall, the Lyapunov Spectrum serves as a useful tool for understanding how network pruning and re-training affect the dynamics of neural networks, which is important for optimizing neural networks for better performance and efficiency.
>
>
> We can get an idea of the principal dynamic characteristic of LNP-HRSNN from the Lyapunov spectrum of the randomly initialized network with a very low probability of connection (p=0.001). We see that such networks were more unstable as the largest Lyapunov exponent was positive. This unstable behavior might be because randomly generated networks lack structured connections that might otherwise guide or constrain the flow of neural activity. Without these structures, the network is more likely to exhibit erratic behavior as the activity patterns are less predictable and more susceptible to amplification of small perturbations, making them more unstable.
>
> [Lyapunov Spectrum of the HRSNN model in three stages of pruning and training and different levels of sparsity for initialization: (a) the Lyapunov Spectrum of unpruned HRSNN model with a probability of connection $p=0.1, 0.01, 0.001$ (b) the Lyapunov Spectrum after LNP pruning (c) the Lyapunov Spectrum after training of the pruned model](https://anonymous.4open.science/r/ICLR24-1FD2/lyapunov_updated.png)
>
>
> ## References:
>
> [1] Engelken, R., Wolf, F. and Abbott, L.F., 2023. Lyapunov spectra of chaotic recurrent neural networks. Physical Review Research, 5(4), p.043044.
>
> [2] Vogt, R., Puelma Touzel, M., Shlizerman, E. and Lajoie, G., 2022. On lyapunov exponents for rnns: Understanding information propagation using dynamical systems tools. Frontiers in Applied Mathematics and Statistics, 8, p.818799.

---

> ### Comment · Reviewer_udA5 · 2023-11-23
> **Thank you for your comment**
>
> I appreciate the authors' thorough and well-structured rebuttal, which has addressed my concerns effectively. It is evident that they have made solid progress in refining their work and providing convincing arguments to support their claims. Taking these revisions into account, I am inclined to raise the paper's score by 1.

---

### Official Review · Reviewer_MeNm · 2023-11-06

**Soundness:** 3 good
**Presentation:** 3 good
**Contribution:** 3 good
**Rating:** 6
**Confidence:** 3

**Summary:**

This paper proposes a novel Lyapunov noise pruning algorithm to design a sparse RSNN from an untrained RSNN. The method is task-agnostic, which can be trained later for different down-streaming tasks. The experimental results shows the effectiveness of this method.

**Strengths:**

- The paper is well-written, with very clear illustration on the methods and implementations.
- This paper proposes a novel task-agnostic method for designing sparse RSNN, which is quite novel compared with conventional approach of designing task-dependent pruning way.
- Compare to conventional network pruning methods of RSNN, the proposed method shows a very good generalization ability, proving by several down-streaming tasks’ consistent performance.

**Weaknesses:**

- The motivation is not fully explained, see the questions below.
- Though in the experimental parts, it is showed to be advantageous compared other pruning methods, overall, it is still hard to see how useful this method can be in real cases. Namely, one might ask, compared to a SOTA method (not only RSNN) in a specific problem (say tasks listed in the experimental parts), how much gain (in terms of performance or efficiency) does this method bring?

**Questions:**

1.	Why focusing on recurrent SNN? I understand Lyapunov noise pruning (LNP) is originally developed for RNNs, why do you choose to consider this method in spiking cases?
2.	RSNN has two hierarchical dynamics, one is the recurrency of network, the other one is the intrinsic recurrency of spiking neurons, do you distinguish these two dynamics? Or on the other hand, whether this method can be used directly in spiking neural networks?
3.	What does the word “heterogeneous” mean here? Is it just the time constant is variable compared with conventional neuron models? (though in many literatures it is already considered)

---

> ### Author Response · Authors · 2023-11-20
> **Response to Reviewer MeNm (1/3)**
>
> ___
> ___
> We thank the reviewer for their insightful comments and constructive feedback on our manuscript. We appreciate the opportunity to clarify and expand upon key aspects of our work. We hope we have addressed their concerns and questions regarding the paper and hope they will reconsider their rating.
> ___
> ___
>
> > The motivation is not fully explained, see the questions below
>
> **>>**  Spiking neural networks (SNNs), employ binary spikes for neuron communication, offering a significant energy-saving advantage in neuromorphic hardware deployment due to their event-driven operations. Recently, there has been a growing interest in developing recurrent SNN architectures with heterogeneous timescales for solving several spatio-temporal classification and temporal prediction tasks. Though introducing such heterogeneities helps improve the model performance, it exponentially increases the computational complexity of the model and makes it extremely hard to train and stabilize.
>
> To address this issue, researchers have been working on developing pruning methods to  significantly reduce the number of neurons and synaptic connections with minimal impact on performance. However, most existing pruning methods, which were initially developed for ANNs, do not fully exploit the unique characteristics of SNNs, particularly the potential for fine-grained neuron pruning. This oversight limits the full realization of the sparse computing capabilities of neuromorphic hardware. Existing pruning works have primarily been adapted from pruning methods designed for ANNs and focused more on enabling SNNs to run on hardware with limited computational resources. Consequently, these methods do not consider the stability of the model for their pruning process, which is essential for modeling a generalizable yet robust model. Moreover, all the current methods aim to optimize performance on a given task during pruning, leading to overfitting to that task. These oversights hinder the full utilization of the sparse computing capabilities and fail to leverage the heterogeneity in dynamics of the RSNN models. These problems motivate the development of new pruning methods for SNNs, particularly for RSNNs.
>
> In this paper, we propose a novel, comprehensive \textbf{task-agnostic method} pruning framework for designing sparse heterogeneous SNNs. This approach capitalizes on the overparameterization in timescales that arises from the heterogeneity in the parameters, and the sparse, event-driven nature of spiking networks. Instead of minimizing (or constraining) performance loss for a given task, our proposed Lyapunov Noise Pruning (LNP) algorithm optimizes the model structure and parameters while pruning to preserve the stability of the sparse HRSNN. This results in sparse models that are stable and flexible, and maintain performance across multiple tasks.
>
> > Though in the experimental parts, it is showed to be advantageous compared other pruning methods, overall, it is still hard to see how useful this method can be in real cases. Namely, one might ask, compared to a SOTA method (not only RSNN) in a specific problem (say tasks listed in the experimental parts), how much gain (in terms of performance or efficiency) does this method bring?
>
> **>>**  We modified Table 1 and reported the baseline accuracy of the unpruned network, the accuracy loss using the pruned network, and the neuron and synapse sparsity levels of the pruned network.
>
> We have also introduced the notion of VPT/FLOPS as a measure of the efficiency of the RSNN models. VPT (Valid Prediction Time) is the amount of time in the future the model can predict within some fixed bounded RMSE (0.1 for our case). So, VPT/FLOPS gives a measure of the efficiency of the model. This result is shown in Table 6 and Figure 6(a) in Suppl Sec B.1
>
> Further, we have directly compared the end “pruned” models (instead of the pruning performance), in terms of (1) accuracy and (2) computational cost, as shown in the attached figure. The computational cost is estimated using Synaptic Operations (SOP), as introduced by Furber et al. For accuracy, we considered classification performance for the CIFAR10 and CIFAR100 datasets. Note, that most of the prior works have reported CIFAR10 results; only very few papers reported CiFAR100.
>
> We also added the notion of average Synaptic Operations (SOP) between the unpruned and pruned network as previously discussed by Furber et al. We also plotted the average SOPs vs absolute performance for the pruned network and compared it in a scatter plot (Fig attached) for the (a) CIFAR10 (b) CIFAR100, and (c) Lorenz63 datasets. The detailed discussion and figures are added in Suppl Sec. B.4
>
> [Scatter Plots for Performance vs SOPs](https://anonymous.4open.science/r/ICLR24-1FD2/scatters.png)

---

> ### Author Response · Authors · 2023-11-20
> **Response to Reviewer MeNm (2/3)**
>
> > Why focusing on recurrent SNN? I understand Lyapunov noise pruning (LNP) is originally developed for RNNs, why do you choose to consider this method in spiking cases?
>
> **>>**  We would like to thank the reviewer for their feedback.
> In this paper, we primarily look into spiking neural network as compared to standard DNNs, SNNs have the energy saving advantage due to their event-driven nature when deployed on neuromorphic hardware. Recently, there has been a growing interest in developing recurrent SNN architectures with heterogeneous timescales, which not only exponentially increases the computational complexity of the model but also makes them hard to train. As such, in this paper, we wanted to leverage these overparameterization in timescales arising due to the heterogeneities and design a much smaller network where the bases for the timescales are orthogonal, helping us prune the network. These optimizations based on heterogeneity in timescales are possible only in spiking neurons, which have the added temporal dimension inherent to them.
>
> We have also extended the algorithm for use on Feedforward Spiking Neural Networks like Spiking ResNets. The results are added in Table 1 and Fig 3. We have also added the description of the algorithm for the Feedforward Spiking networks in Supplementary Section A.2 . A short description of the changes for the feedforward SNN is given below:
>
> 1.	For the structured pruning in Step 2, instead of pruning neurons, we prune channels
> 2.	For Step 3, we use norm preservation using skip connections
>
> We use the following terminology for the different pruning cases:
>
> a.	*Randomly Initialized SNN*: The SNN model is not trained before pruning, and the parameters are randomly initialized. Only the final pruned network is then trained.
> b.	*Converted SNN*: A standard DNN was trained on a dataset and then converted to an SNN with the same architecture using DNN-to-SNN conversion methods.
>
> ### Table showing the performance of LNP on Feedforward Spiking Neural Networks
> | **Model** | **Baseline Accuracy (CIFAR10)** | **Accuracy (CIFAR10)** | **Neuron Sparsity (CIFAR10)** | **Synapse Sparsity (CIFAR10)** | **Baseline Accuracy (CIFAR100)** | **Accuracy (CIFAR100)** | **Neuron Sparsity (CIFAR100)** | **Synapse Sparsity (CIFAR100)** |
> |-----------|---------------------------------|------------------------|-------------------------------|---------------------------------|----------------------------------|-------------------------|--------------------------------|----------------------------------|
> | Spiking ResNet19 (Randomly Initialized) | 92.11±0.9 | -2.15±0.19 | 90.48 | 94.32 | 73.32±0.81 | -3.56±0.39 | 90.48 | 94.32 |
> | Spiking ResNet19* (converted) | 93.29±0.74 | -0.04±0.01 | 93.67 | 98.19 | 74.66±0.65 | -0.11±0.02 | 94.44 | 98.07 |
>
> The results are added in Table 1 and Fig 3. We have also added the description of the algorithm for the Feedforward Spiking networks in Supplementary Section B.2. The results for the Spiking ResNet models are given in Table 7 in Supplementary Section B.2
>
>
>
>
>
> > RSNN has two hierarchical dynamics, one is the recurrency of network, the other one is the intrinsic recurrency of spiking neurons, do you distinguish these two dynamics?
>
> **>>**  Our pruning approach distinguishes between the recurrent connection in the neural network and inherent recurrency (i.e., time-dependent evolution of membrane potential) within a spiking neuron. Steps 1,2,3, where we prune the network synapses and neurons and then regenerate some synapses for eigenvector preservation, consider the recurrent connections in the network. On the other hand, Step 4, where we optimize the neuronal timescales, considers the recurrency within the spiking neuron and modulates the time-dependent recurrent dynamics of the neurons.
>
> >  Or on the other hand, whether this method can be used directly in spiking neural networks?
>
> **>>**   We would like to thank the reviewer for pointing this out. We have extended the algorithm for use on Feedforward Spiking Neural Networks like Spiking ResNets. The results are added in Table 1 and Fig 3. We have also added the description of the algorithm for the Feedforward Spiking networks in Supplementary Section A.2 .

---

> ### Author Response · Authors · 2023-11-20
> **Response to Reviewer MeNm (3/3)**
>
> > What does the word “heterogeneous” mean here? Is it just the time constant is variable compared with conventional neuron models? (though in many literatures it is already considered)
>
> **>>**  The reviewer is correct. The heterogeneity implies that different neurons in the network has different time constants (obtained from a distribution) resulting in different integration and relaxation dynamics. We also that heterogeneity of neurons has been shown in previous papers, some of which have been cited in the paper [1-3].
> We would like to stress that the heterogeneity in the timescales is not the main contribution of the paper. The focus of this paper is to develop a pruning method that leverages and tunes heterogeneity to engineer a sparser (fewer neurons and synapses) SNN network.
>
>
> ## References
> [1] Perez-Nieves, N., Leung, V.C., Dragotti, P.L. and Goodman, D.F., 2021. Neural heterogeneity promotes robust learning. Nature communications, 12(1), p.5791.
>
> [2] Chakraborty, B. and Mukhopadhyay, S., 2023. Heterogeneous recurrent spiking neural network for spatio-temporal classification. Frontiers in Neuroscience, 17, p.994517.
>
> [3] She, X., Dash, S., Kim, D. and Mukhopadhyay, S., 2021. A heterogeneous spiking neural network for unsupervised learning of spatiotemporal patterns. Frontiers in Neuroscience, 14, p.1406.

---

### Author Response · Authors · 2023-11-20
**Summary of Changes**

We thank the reviewers for their constructive feedback and insightful comments. With regard to the reviews, the significant changes in the updated paper are listed as follows:
## Major Changes
1. **Detailed Motivation:** We added a more detailed motivation for our work and a more detailed discussion on the need for task-agnostic pruning methods
2. **LNP Algorithm for Spiking Feedforward Network:** We extended our LNP algorithm for the Spiking Feedforward Network and tested the algorithm on the Spiking ResNet on more difficult CIFAR10 and CIFAR100 datasets. We also repeated the experiment multiple times to report the mean and standard deviation of the results.
3. **Synaptic Operations:** We used Synaptic Operations to calculate the efficiency of the spiking neural network models.
4. **Updated Comparison Metric:** We updated the comparison metric to include Accuracy loss, Efficiency gain (using the ratio of synaptic operations), and neuron and synapse sparsity. We compared it with current state-of-the-art pruning methods.
5. **Calculation of Sparsity:** We updated the calculation of the sparsity metrics by including the readout layers – we have also added a discussion on how the readout layer is being implicitly pruned
6. **Generalizability Study of LNP Pruned Model:** Investigated the generalizability of the LNP model by comparing its performance with other pre-trained networks that are overfitted to the dataset used for pruning.
7. **Dynamic Characteristics Study:** Analyzed the dynamic characteristics of the model using the Lyapunov Spectrum of the RSNN.
8. **Lyapunov Spectrum Computation Discussion:** Added a more detailed discussion on the computation of the Lyapunov Spectrum.
9. **Description of Linearization Method:** Expanded the description of the linearization method around critical points.
10. **Activity-Based Pruning Baseline Description:** Provided a more comprehensive description of the Activity-based pruning baseline used in the study.
11. **Comparison with Random Sparse Initializations:** Included a comparison with Random sparse initializations, demonstrating how such methods result in unstable models.


## New Results and Figures:
1. **Scatter Plots:** We added scatter plots showing the Avg. Synaptic Operations (SOPs) vs Accuracy for CIFAR10, CIFAR100, and Lorenz63 datasets.
2. **Table 1 Updates:** We made the results in Table 1 more extensive and complete with more baseline pruning methods. We tested on more complex datasets like CIFAR10 and CIFAR100 and repeated each experiment 10 times to report the mean and standard deviation. We also compared the performance of our pruning method with other state-of-the-art methods. The comparison focused on Baseline accuracy, accuracy loss, the average SOP ratio between unpruned and pruned networks, Neuron and Synapse Sparsity for both CIFAR10 and CIFAR100 datasets.
3. **Dynamic Characteristics using Lyapunov Spectrum:** We plotted the Lyapunov spectrum of the HRSNN model before pruning, after pruning, and after training to study the dynamical characteristics of the model.
4. **Ablation Study:** We conducted an ablation study to examine the differential effect of each of the four steps of the LNP algorithm.
5. **Random Sparse Initialization Performance:** We updated Fig 3 (c), which showed the change in RMSE vs iterations for the LNP pruning method on the HRSNN model. We included the result when we randomly generated a network with an equal number of synapses and edges at each step of the iteration and showed that this random network is much more unstable and exhibits worse average performance than LNP.

## Other Changes

1. **Updated Citations:** We updated the citations to avoid repetition.
2. **Improved Readability:** We updated the text to enhance readability.
3. **Detailed Discussion on HRSNN Architecture:** Added a comprehensive discussion on the HRSNN architecture and its training/testing methods.
4. **Lyapunov Spectrum Calculation Discussion:** Expanded the discussion on the calculation of the Lyapunov spectrum of the spiking RSNN.
5. **Clarification of Mathematical Model:** Added further clarification and a detailed description of the mathematical model for the noise-based synapse pruning method, enhancing readability.
6. **Lorenz and Rossler Datasets:** Included additional discussions on the Lorenz and Rossler datasets used for the prediction task.
7. **Block Diagram for Methodology:** Added Figure 6 in Supplementary Section A.6, showcasing the block diagram of the methodology using HRSNN for prediction/classification tasks.
8. **Updated Figure in Supplementary Section:** Revised Fig. 8 in Supplementary Section B.1 to display the bar graph of VPT/SOPs for various datasets.

We hope we have addressed the reviewers’ concerns and questions regarding the paper and that they will reconsider their rating based on these changes.

---

### Author Response · Authors · 2023-11-22
**Request for Reconsideration of Rating**

We sincerely thank the reviewer for their valuable and insightful feedback on our manuscript. Your comments have provided us with a unique opportunity to enhance and clarify vital elements of our research. In response, we have meticulously revised our manuscript, ensuring that all concerns and queries raised have been thoroughly addressed to the best of our abilities.

We believe these modifications have significantly strengthened the quality and clarity of our work. Therefore, we kindly request you to reevaluate our manuscript in light of these improvements and reconsider and increase your initial rating. We are confident that these revisions will positively impact your assessment and look forward to your feedback.

---

### Meta-Review · Area_Chair_2zda · 2023-12-11

**Metareview:**

This paper generated considerable discussion amongst the reviewers, and I congratulate the authors on their detailed and comprehensive rebuttals, which were critical in leading some reviewers to raise their scores. Ultimately, 3/4 reviewers scored the paper as above threshold for acceptance, and after reading their comments I am inclined to agree with this majority assessment.  I'm pleased to report that this paper has been accepted to ICLR.  Please revise the manuscript to address all reviewer comments and questions.

**Justification For Why Not Higher Score:**

Two reviewers gave very high scores of 8, the other two (5,6) were much closer to the threshold.  It seems to me that a poster accept is the right decision, based on the level of overall enthusiasm, but I could be persuaded to bump it up to a spotlight.

**Justification For Why Not Lower Score:**

Two reviewers gave very high scores of 8, outweighing two other reviewers who scored it close to threshold.  It seems the work is solid and interesting enough to warrant acceptance, in my opinion.

---

### Decision · Program_Chairs · 2024-01-16

Accept (poster)